# Expanded iOn switch toolkit enables flexible clonal labeling and dynamic imaging in model and non-model animals
Zi Chao Ngiam[1,2,9], Kyosuke Wada[3,4,9], Jun Hatakeyama[5], Yuki Y. Yamauchi[6], Takuya Kaneko[7,8], Pauline Rouillard[6], Haruka Sato[5], Masahiko Hibi [7], Ikuo K. Suzuki [6], Carina Hanashima [1,2], Chiaki Ohtaka-Maruyama [3] & Takuma Kumamoto [3] ✉

Understanding how neural progenitors generate diverse neuronal subtypes is central to developmental and evolutionary neuroscience. Lineage tracing is a key tool for dissecting these processes, but existing methods are often constrained by stochastic expression, recombination bias, or reliance on transgenic models—limitations that restrict their application in non-model organisms. Here, we present an enhanced iOn switch system optimized for evo-devo research. Leveraging its modular, integration-based transgene expression logic, we developed a tunable labeling strategy that enables both sparse and dense labeling within a single framework. We further expanded the system's fluorescent palette and introduced subcellular targeting options to support multiclonal analysis and stable time-lapse imaging. Finally, we demonstrate the versatility of this approach across a wide range of vertebrate species, including mouse, guinea pig, rat, chick, turtle, and zebrafish, thereby establishing the iOn switch system as a flexible and accessible tool for evo-devo research.

Understanding how the brain develops—from progenitor behavior to mature circuit formation—requires the integration of molecular, cellular, and systems-level insights. While much of our foundational knowledge has come from genetically tractable models such as the mouse, comparative developmental studies across vertebrate lineages offer insights into how structurally distinct yet functionally analogous brains emerge through evolution[1–4]. Among the various levels at which cortical development can be examined, the clonal level offers a uniquely informative scale for comparative analysis, as each clone represents the basic developmental unit from which cortical architecture is assembled. By resolving the output of individual progenitors, clonal analysis enables precise quantification of lineage size, fate potential, migratory behavior, and spatial organization—key parameters for identifying conserved mechanisms and evolutionary innovations across species[5–7]. Critically, these analyses necessitate lineage tracing tools that can reproducibly label and track single progenitor-derived

lineages across different animal model systems. However, lineage tracing tools suitable for comparative research remain limited. Many established methods rely on transgenic models or recombinase-based systems, restricting their use to a small subset of genetically accessible organisms[8,9]. Some approaches require extensive optimization across developmental contexts and species[10]. Others, such as promoter- and enhancer-driven systems, often rely on species-specific regulatory elements that may not yield comparable expression patterns when broadly applied, limiting their generalizability[11].

The iOn switch system was originally developed to overcome several of these limitations by coupling reporter expression strictly to genomic integration via the PiggyBac transposase (PBase), thereby eliminating transient episomal expression without the need for secondary silencing steps[12]. Its compatibility with electroporation-based delivery enables rapid, transgene-independent deployment across a wide range of embryonic models,

[1]Department of Biology, Faculty of Education and Integrated Arts and Sciences, Waseda University, Tokyo, Japan. [2]Graduate School of Advanced Science and Engineering, Waseda University, Tokyo, Japan. [3]Developmental Neuroscience Project, Department of Basic Medical Sciences, Tokyo Metropolitan Institute of Medical Science, Tokyo, Japan. [4]Developmental Neurology, Molecular and Cellular Medicine Course, Niigata University Graduate School of Medical and Dental Sciences, Niigata, Japan. [5]Institute of Molecular Embryology and Genetics, Kumamoto University, 2-2-1 Honjo, Chuo-ku, Kumamoto, Japan. [6]Department of Biological Sciences, Graduate School of Science, The University of Tokyo, Tokyo, Japan. [7]Department of Biological Science, Graduate School of Science, Nagoya University, Aichi Nagoya, Japan. [8]Institute for Advanced Research, Nagoya University, Aichi Nagoya, Japan. [9]These authors contributed equally: Zi Chao Ngiam, Kyosuke Wada. ✉e-mail: kumamoto-tk@igakuken.or.jp

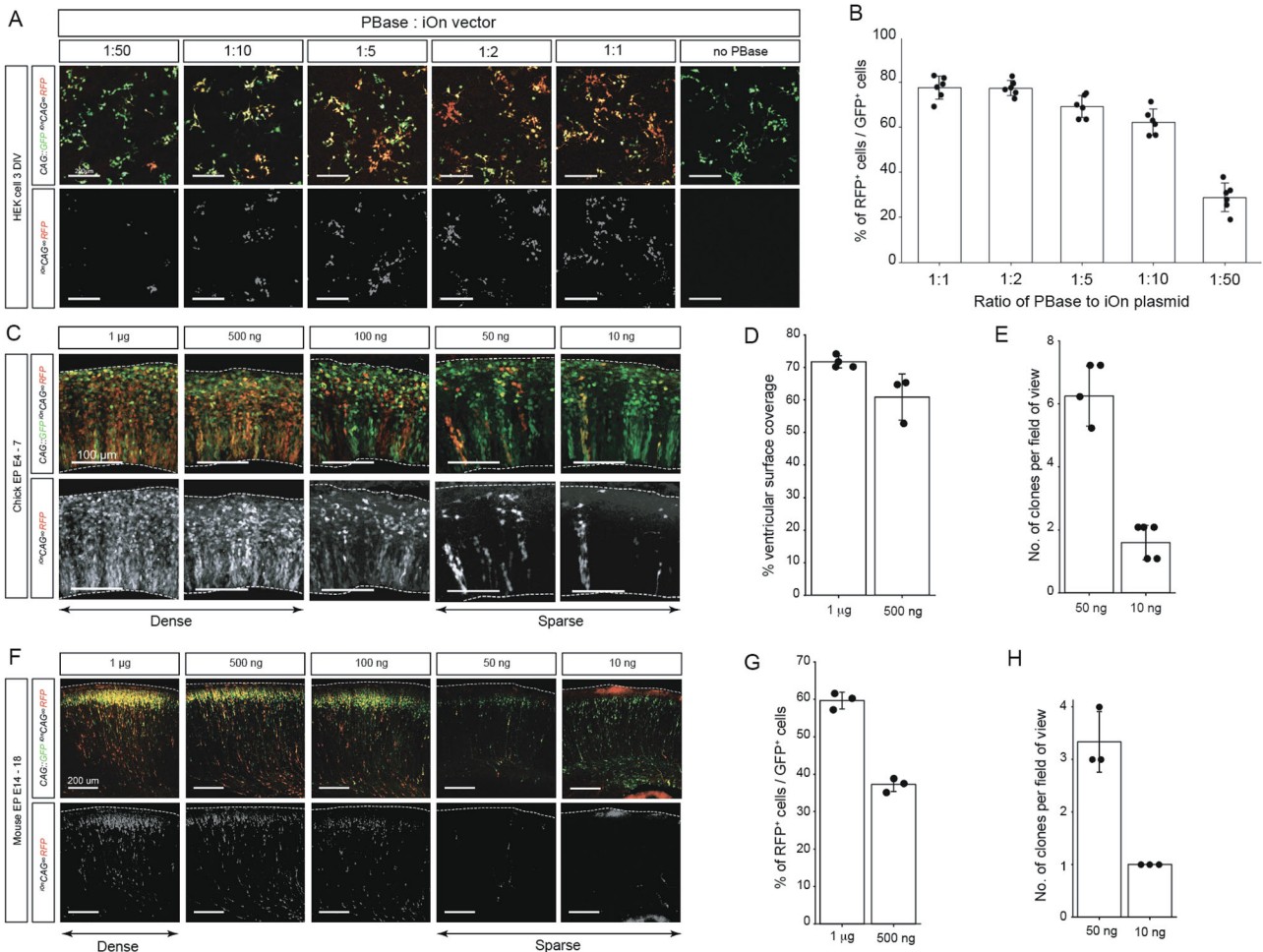

**Fig. 1 | Optimization of piggyBac vs iOn plasmid ratio and tunable control of labeling density across species. A** Representative images of HEK293 cells co-transfected with *CAG::GFP* and *iOnCAG ∞ RFP* plasmids using varying PBase:iOn plasmid ratios. **B** Quantification of iOn labeling within the transfected population, expressed as the fraction of RFP⁺ cells among GFP⁺ cells (RFP⁺/GFP⁺) at each PBa-se:iOn ratio. Each transfection was treated as one independent replicate; n = 6 per condition (2 independent experimental repeats × 3 transfections per condition). Multiple fields within a well were averaged to a single value per transfection. **C** Representative images of the ventricular zone in chick embryos electroporated at E4 and analyzed at E7 using a range of donor plasmid concentrations (fixed PBa-se:iOn ratio). Quantification of labeling in chick brains: **D** Estimated ventricular-surface labeling coverage under dense conditions (1 μg and 500 ng); **E** Under sparse conditions (50 ng and 10 ng), labeling was quantified as the estimated number of labeled clones per brain. Because the color palette is limited, clone numbers were estimated conservatively by treating each contiguous multicellular cluster (>3 labeled cells) as one putative clone; single cells and two-cell clusters were not counted as clones. Each embryo was treated as one independent replicate. Group sizes by donor concentration: 1 μg: *n* = 4; 0.5 μg: *n* = 3; 0.05 μg: *n* = 2; 0.01 μg: *n* = 5. For each embryo, FoVs within sections were averaged to yield a single value. **F** Coronal brain

sections from mouse embryos electroporated at E14 with *CAG::GFP* and *iOnCAG ∞ RFP* at varying plasmid concentrations, using a fixed PBase:iOn ratio of 1:5. Brains were analyzed at E18. Quantification of labeling in mice brain: **G** RFP + / GFP+ cell ratio (1 μg); **H** Estimated number of labeled clones per field of view under sparse conditions (50 ng and 10 ng). Clone counts were estimated conservatively as above (contiguous multicellular clusters >3 labeled cells; single cells and two-cell clusters excluded). Each embryo was treated as one independent replicate. Group sizes by donor concentration: 1 μg: *n* = 3; 0.5 μg: *n* = 3; 0.05 μg: *n* = 3; 0.01 μg: *n* = 3. For each embryo, FoVs within sections were averaged to yield a single value. "1 μg" indicates a 1 μg/μL DNA solution, and 1 μL was injected per embryo (i.e., 1 μg DNA per embryo). Error bars indicate mean ± s.e.m. Because non-electroporated cells are unlabeled and therefore not visible in our fields, the RFP⁺/GFP⁺ metric reports the relative fraction of iOn-labeled cells within the electroporated population, rather than the absolute fraction of all cells in the tissue. Input ratios were set on a molar (copy-number) basis where applicable, with weights provided for reference. When PBase was supplied as mRNA (2,107 nt), the PBase mRNA:iOn condition corresponds to ~0.96:1 by molecules (iOn plasmid 1.0 μg, 5223 bp → 0.290 pmol; PBase mRNA 0.2 μg, 2,107 nt → 0.279 pmol), previously referred to as 1:5 by weight. Clone calling was conservative due to the limited color palette (see (**E**) and (**H**)).

including both in utero and in ovo systems[13]. Here, we present an expanded and systematically optimized version of the iOn-switch toolkit tailored for comparative and evolutionary developmental (evo-devo) studies. We evaluate its performance across several domains: optimizing PBase-to-donor ratios for efficient genomic integration, introducing tunable plasmid concentrations to modulate labeling density, expanding the fluorescent and subcellular labeling toolkit, and assessing its applicability across multiple vertebrate species. These improvements enable reproducible clonal labeling in both sparse and dense contexts, support compatibility with live-cell imaging, and facilitate downstream computational analysis across diverse animal model systems.

## Results

### Optimization of PBase to iOn Plasmid Ratio

To maximize integration efficiency while minimizing cytotoxicity, we first optimized the ratio of PBase to iOn plasmid. Although PBase is generally not subject to overproduction inhibition[14], elevated levels of transposase and donor plasmid have been associated with toxicity and mosaic expression patterns in vivo[15]. We co-transfected HEK293 cells with 1 μg of control episomal *CAG::GFP* and *iOnCAG ∞ RFP* plasmids along with varying amounts of PBase plasmid (Fig. 1A). Quantification of RFP⁺/GFP⁺ cell ratios revealed that a minimum 1:10 ratio was required to exceed 60% efficiency, with limited improvement beyond 1:5 (Fig. 1B). We selected the 1:5 ratio for

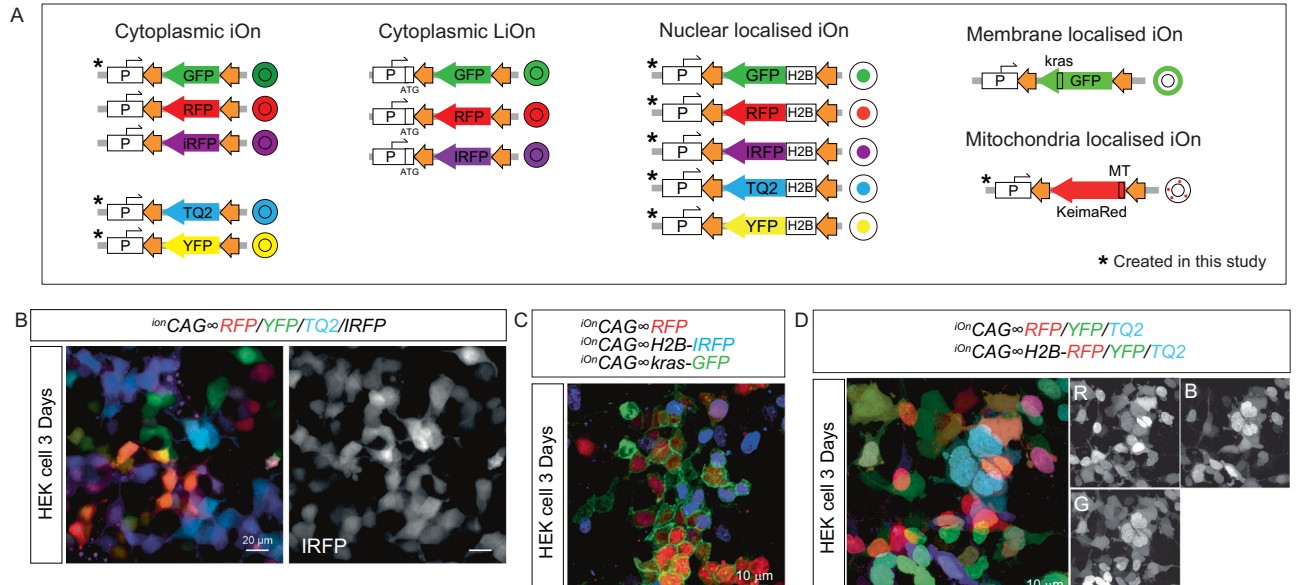

**Fig. 2 | Expansion of the iOn switch toolkit for multicolor and subcellular labeling. A** Schematic overview of all available iOn switch plasmids, including both previously established and newly developed constructs categorized by fluorophore and subcellular targeting. **B** HEK293 cells labelled with four-color iOn constructs (*iOnCAG ∞ RFP*, *-YFP*, *-TQ2*, and *-iRFP*); images shown 3 days post-transfection. **C** Compartment-specific labeling using a combination of *iOnCAG ∞ RFP* (cytoplasm), *-H2B-iRFP* (nucleus), and *-KRAS-GFP* (membrane). **D** Example of labeling three-color iOn constructs with different localization tags (cytoplasmic and nuclear-targeted variants of *RFP*, *-YFP*, and *-TQ2*), demonstrating the combinatorial expansion possible within the iOn switch system.

downstream experiments based on its consistent performance and alignment with previously reported recommendations[16].

## Tuning labeling density across species

We next asked whether the concentration of iOn plasmid could be modulated to control labeling density while keeping the PBase: iOn ratio constant. This tunability is particularly valuable as it would allow the iOn switch system to be used for both experimental designs requiring sparse labeling for single-clone analysis and dense labeling for multi-color clonal reconstruction. We first tested the effect of varying plasmid concentration on labeling density in the chick pallium. Electroporation was performed at E4 under electroporation conditions of 15 V with one square pulse (50 ms on, 950 ms off) by co-electroporating 1 μg of control *CAG::GFP* and *iOnCAG ∞ RFP* at varying concentrations (Fig. 1C). Brains were collected three days later, and labeling was assessed either by estimating labeling coverage in the ventricular zone (Fig. 1D; area-based proxy) or by quantifying relative labeling density within the electroporated population (Fig. 1E; e.g., RFP+/GFP+ among *CAG::GFP*-positive cells) and counting labeled clones per field of view. 1 μg and 500 ng yielded high-density labeling of 70% and 60% coverage, respectively, while 50 ng and 10 ng produced sparse labeling patterns of 6 and 1.5 clones per field of view, respectively (Fig. 1C–E). To test whether this relationship holds across species, we performed clonal analysis using the same concentration in mice. Electroporation was performed at E14 under electroporation conditions of 40 V with four square pulses (50 ms on, 950 ms off), and brains were collected at E18. As in the chick embryos, a similar decrease in labeling density was observed as the plasmid concentration was decreased, with dense labeling achieved with 1 μg, while 10 ng consistently labeled only 1–2 clones per brain (Fig. 1F–H). These results support the idea that labeling density can be reliably modulated by plasmid concentration. Moreover, comparing labeling efficiencies at 1 μg across species revealed that both the chick and mouse brains exhibited similar outcomes, consistent with observations in HEK cells. This strengthens the conclusion that the iOn system operates effectively across different species and is not constrained by model-specific differences.

## Expansion of fluorescent and subcellular labeling capabilities

While the original iOn switch system provided reliable labeling, its limited fluorescent palette constrained its use in multi-clonal experiments. To overcome this, we introduced the mTurquoise2 (TQ2) and Yellow Fluorescent Protein (YFP) to complement the existing RFP reporter. This enables tricolor combinations for multi-color labeling, while keeping the infrared fluorescent protein (iRFP) channel available for future immunostaining or as a fourth fluorescent readout. To further expand labeling dimensionality, we created iOn constructs targeted to specific subcellular compartments, including the nucleus, membrane, and mitochondria. All constructs were validated in HEK293 cells to confirm correct localization and robust expression (Fig. S1A–C). A comprehensive schematic of all current and newly developed iOn switch plasmids is provided in Fig. 2A. We also demonstrate the versatility of these tools in HEK293 cells through several representative applications: four-color cytoplasmic labeling using *iOnCAG ∞ RFP*, *-YFP*, *-TQ2*, and *-iRFP* (Fig. 2B); compartment-specific labeling using *iOnCAG ∞ RFP*, nuclear *-H2B-iRFP*, and membrane-localized *-KRAS-GFP* (Fig. 2C); and a three-color strategy combining cytoplasmic and nuclear-targeted variants of *-RFP, -YFP*, and *-TQ2* (Fig. 2D). Notably, when nuclear and cytoplasmic iOn switch plasmids are co-transfected, the nuclear variants exhibit stronger fluorescence intensity which can overpower cytoplasmic signals in the same cell. To balance signal detection across compartments, we recommend adjusting the nuclear to cytoplasmic plasmid ratio to 1:3. Together, these tools greatly expand the labeling capacity of the iOn switch system, enabling more complex lineage tracing by facilitating both manual and software-assisted segmentation.

## Clonal Reconstruction and multicolor labeling in Mouse and Non-Model animals

We next tested whether the updated iOn switch system could facilitate clonal reconstruction in vivo. Using optimized dense-labeling conditions, we electroporated chick dorsal pallium at E4 and analyzed the tissues at E6, and mouse cortices at E14 and analyzed the tissues at P7. Fluorescence intensity from each channel was quantified for individual cells, and relative expression ratios were plotted on ternary plots. Clustering analysis of these

plots enabled retrospective inference of putative clones based on shared fluorescence signatures and validated through spatial proximity (Fig. 3A–D). To better define the clonal analysis strategy under high labeling density, we used a densely labeled chick pallium dataset as an example. Specifically, we plotted single-cell red/green/blue contributions in ternary space and incorporated spatial context to distinguish adjacent clones that partially overlap in color space (Fig. S3). Moreover, under multicolor labeling conditions, we observed no obvious increase in cytotoxicity despite introducing multiple iOn plasmids into the tissue (Fig. S4, Supplementary Movie 1).

To assess whether the iOn toolkit generalizes beyond standard model organisms, we extended our experiments to chick[12], turtle[17], rat, guinea pig[18], and zebrafish[19,20] to demonstrate multicolor labeling under species-appropriate delivery conditions (Figs. 3A and 4; S5, S6). In turtle, rat, and guinea pig, iOn plasmids were delivered into the embryonic cortical ventricle by electroporation, whereas in zebrafish embryos iOn plasmids were delivered by microinjection. In turtle, rat, and guinea pig, iOn robustly labeled cortical projection neurons and glia, which could be distinguished based on their morphology (Fig. 4A–C). In zebrafish, multicolor labeling was readily detected in spinal neurons at 1 dpf (Fig. S6), while iOn switch expression became prominent in muscle after 5 dpf (Figs. 4D; S6). This tissue- and stage-dependent expression pattern in zebrafish may reflect differences in promoter usage (*ubiquitin B*, *pUbb*, in zebrafish versus *pCAG* in other species), species-dependent piggyBac integration efficiency, and/or the delivery method (microinjection versus electroporation). Clear GFP fluorescence was observed in more than 60% of injected animals. In approximately 10% of injected animals, muscle labeling was detected throughout the trunk, as shown in Fig. 4D. However, it should be noted that not all color plasmids are incorporated and expressed in every case. Even when the same delivery procedure is applied within the same species, occasional skewed expression can occur; for example, in rare cases in turtle we observed markedly reduced GFP expression (Fig. S5). The underlying cause of this variability remains unclear. Therefore, especially when applying the system to a new species, empirical optimization of delivery conditions is recommended. Collectively, these results indicate that iOn is broadly applicable for multicolor labeling across diverse developing tissues in non-model animals.

### Live imaging with stable fluorescent expression

The iOn switch system's integration-coupled expression provides a key advantage for live imaging applications, where consistent fluorescence intensity is critical. This contrasts with traditional labeling methods using *CAG* promoters, often resulting in highly variable expression due to differences in plasmid copy number and promoter activity, complicating time-lapse imaging. To directly compare these systems in a live imaging context, we electroporated mouse embryos at E14 with both *CAG::GFP* and *iOnCAG ∞ RFP*, dissected the brains at E16, and performed two days of slice-based time-lapse imaging. During imaging of the migratory trajectory from the ventricular zone (VZ) to the cortical plate (CP), GFP-labeled cells exhibited overexposure near the electroporation site and underexposure in more distant regions, making it difficult to apply a single exposure setting. In contrast, RFP cells displayed uniform fluorescence, enabling stable visualization across the field without requiring dynamic exposure adjustments. This stability facilitated uninterrupted tracking of migrating cells from their origin to their destination (Fig. 5A, B, Supplementary Movie 2). Similar results were obtained in chick embryos (Fig. S2), supporting the system's broader utility for dynamic developmental imaging.

### Testing mRNA delivery to reduce expression latency

Despite its advantages, one limitation of the iOn switch system is the delayed onset of fluorescence. This is likely due to the sequential steps required for PBase transposase expression, genomic integration, and reporter activation. We hypothesized that delivering *PBase* as mRNA, rather than plasmid DNA, could reduce this latency by bypassing transcription. To identify *PBase* mRNA:iOn ratios that would yield comparable transfection efficiencies to DNA delivery, we co-transfected HEK293 cells with *CAG::GFP* and *iOnCAG ∞ RFP*, along with *PBase* mRNA at varying ratios. Transfection efficiency was assessed by calculating the ratio of RFP⁺ to GFP⁺ cells. Unlike PBase DNA, where efficient integration was observed as low as 1:10, only the higher *PBase* mRNA ratios (1:2 and 1:1) approached similar levels of efficiency of around 50% (Fig. S7A and B). Note that these ratios are reported by mass; because *PBase* mRNA and plasmid DNA differ in length, identical mass ratios do not correspond to identical copy-number (molar) ratios. With iOn fixed at 100 ng, our *PBase* mRNA titration series (100, 50, 20, 10, and 5 ng) corresponds to molar ratios (*PBase* mRNA:iOn DNA) of ~4.81:1, 2.41:1, 0.962:1 ( ~ 1:1), 0.481:1, and 0.241:1, respectively. We then compared these two mRNA conditions to the standard 1:10 PBase DNA condition in a time-lapse imaging assay, monitoring RFP expression as a proxy for activation onset. Surprisingly, we observed no significant difference in fluorescence latency between the mRNA and DNA PBase conditions, even at the highest molar ratios tested, suggesting that transposase expression may not be the primary rate-limiting step in the iOn switch system (Fig. S7C, D).

## Discussion

Lineage tracing is a fundamental experimental approach in developmental biology, providing insights into how neural progenitors generate diverse cell types that organize spatially and assemble into functional circuits. However, in non-model organisms, setting up lineage tracing experiments remains technically demanding. Currently available tools like UbC-StarTrack[21], CLoNe[22] or MAGIC markers[23] often require transgenic animals, fixed promoter systems, or include an element of stochasticity, which makes it challenging to implement in many species. For researchers working in comparative or evolutionary developmental biology (Evo-Devo), these limitations create a major barrier to experimentation. The iOn switch system was originally developed to address some of these gaps by offering a modular, plasmid-based, integration-dependent labeling strategy[12]. Here, we present an enhanced version of the iOn switch system, optimized for use in both clonal lineage reconstruction and dynamic live imaging experiments (Fig. 6).

One major challenge in lineage tracing is adapting tools for both sparse single-clone analysis and dense multicolor labeling. In many existing systems, achieving both requires multiple genetic lines or extensive re-optimization. Our updated iOn switch system overcomes this by introducing a simple logic: labeling density can be modulated solely by adjusting the total iOn plasmid concentration, while maintaining a fixed PBase ratio. This makes it easy to shift between sparse and dense conditions without altering the experimental design or underlying vectors. We further show that for multicolor labeling, once the total plasmid concentration is optimized (e.g., 1 µg), it can be evenly partitioned across different fluorophores (e.g., RFP, YFP, TQ2) without compromising transfection efficiency. This "plug-and-play" approach simplifies complex multicolor experiments and eliminates the need for recalibration of each color, an important advantage over existing stochastic or recombinase-based methods. One potential concern is cytotoxicity associated with *in utero* electroporation. Under standard parameters, *in utero* electroporation is generally reported to yield high embryo survival and low cytotoxicity[24,25]. However, electroporation-associated toxicity can increase with plasmid size and/or total DNA load[26]. Accordingly, in this study, we used endotoxin-free plasmid preparations[27], limited the concentration of each plasmid to ~1 µg/µL, minimized the total DNA amount during co-electroporation[25,28], and optimized pulse settings. To directly assess cytotoxicity in our system, we performed (i) time-lapse imaging under high-concentration electroporation conditions and (ii) cleaved caspase-3 immunostaining on brains electroporated under the same conditions. In time-lapse recordings, we rarely observed cells undergoing cell death (Fig. S4A, B, and Supplementary Movie 1), and the cleaved caspase-3 signal was not increased in the electroporated region compared with the surrounding tissue (Fig. S4cb-EP region, compared to Ca-no-EP region). Together, these results suggest that

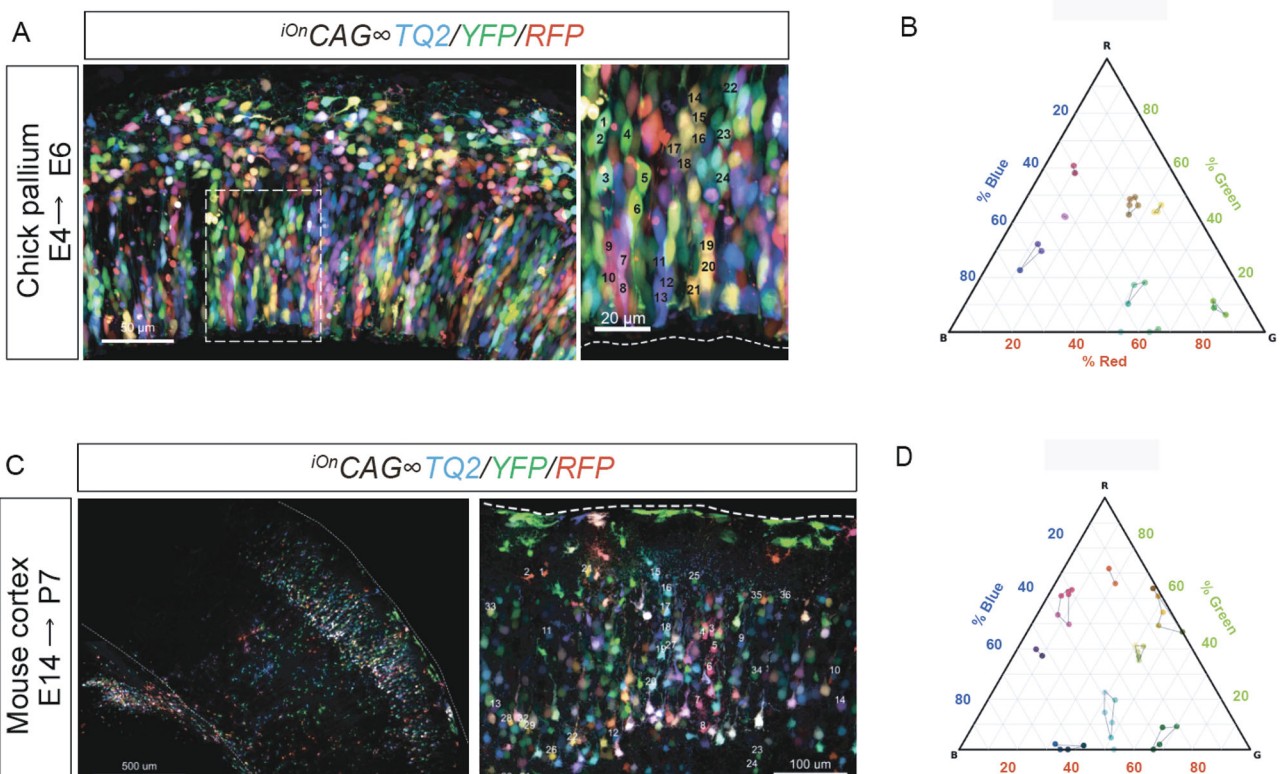

**Fig. 3 | Multicolor clonal reconstruction using the iOn switch system in the chick and mouse cortex. A** Chick brain electroporated at E4 and analyzed at E6 with the three-color iOn constructs (*iOnCAG ∞ RFP*, *-TQ2* and *-YFP*). Cells were blindly assigned identifiers for unbiased lineage analysis without saturation images. **B** Ternary plot showing relative fluorescence of TQ2, YFP, and RFP per cell from (**A**). Clustered profiles indicate putative clones. **C** Representative image of a mouse cortex electroporated at E14 with three-color iOn constructs (*iOnCAG ∞ RFP*, *-TQ2* and *-YFP*) collected at P7. Cells were blindly assigned identifiers for unbiased lineage analysis without saturation images. **D** Ternary plot showing relative fluorescence of TQ2, YFP, and RFP per cell from **C**. Clustered profiles indicate putative clones. Ternary plots display single-cell values for visualization; all statistics (where applicable) are computed at the embryo level to avoid pseudo-replication.

the high-DNA electroporation used for multiplex clonal analysis with the iOn switch induces minimal cytotoxicity in our experimental conditions.

Dynamic processes such as neuronal migration, axon growth, and cell shape changes are central to understanding neural development. However, capturing these events through long-term time-lapse imaging is often limited by signal instability in traditional labeling systems. Plasmids driven by strong promoters like *CAG* can produce highly variable expression levels, leading to overexposure at the electroporation site and underexposure in distant regions, especially during long-range migration.

Our iOn switch system couples reporter expression to genomic integration. In HEK293T cells, we previously quantified integrated iOn cassette copy number and found that it spans ~1.1–6.9 integrated copies under our in vitro conditions[12]. By contrast, we did not measure genomic copy number in the present in vivo experiments in mouse or chick, and therefore, we do not assign a numerical copy-number value to these models. Nevertheless, in vivo piggyBac transgenesis studies show that founders can carry single to multiple integrations, including cases exceeding 10[29], underscoring that integration outcomes can vary by experimental context. Consistent with the expected behavior of integration-coupled expression, our in vivo data show that iOn labeling produces (i) more uniform fluorescence across cells, reflected by a lower coefficient of variation than a conventional *CAG*-driven vector (Fig. 5B), and (ii) sustained, mitotically stable expression that supports long-term clonal tracking during live imaging (Fig. 5A, Supplementary Movie 2). Future work directly quantifying copy number in vivo will be necessary to determine copy-number distributions across in vivo contexts and species.

While the iOn switch system offers robust and stable expression, one limitation remains its delayed onset of fluorescence following electroporation. This latency sets a lower bound on when electroporated cells can first

be visualized and thus constrains experiments targeting rapid developmental processes or short-lived progenitor populations. In contexts such as early lineage bifurcation or transient migratory events, the ability to observe cells soon after targeting is particularly critical. To explore whether this lag could be reduced, we tested whether delivering *PBase* as mRNA (rather than DNA) might accelerate expression by bypassing transcription. Surprisingly, we found no significant difference in onset timing between PBase and DNA conditions, despite comparable transfection efficiencies. This suggests that transposase availability is not the primary bottleneck. Instead, other steps, such as the integration process and subsequent transcription/translation, may contribute more significantly to the delay. In addition, once the reporter protein is produced, fluorophore maturation kinetics could further influence the time to detectable fluorescence. Future iterations of the system could therefore explore fluorophores with faster maturation (e.g., mScarlet-I3)[30] or reporter designs with reduced maturation time to further shorten this window, which would be especially valuable for early lineage studies or short-term live imaging.

A final and critical aim of this study was to demonstrate that the iOn switch system is adaptable across a wide range of vertebrate species, not just traditional genetic models. Building on initial demonstrations in mouse and chick, we extended our validations to include turtle, guinea pig, rat, and zebrafish, and showed that iOn-mediated labeling was consistent and analyzable across all systems. The original paper also demonstrated compatibility with human stem cell-derived tissues emphasizing the evolutionary breadth of its application[12]. This cross-species versatility, combined with the system's modularity and electroporation compatibility, makes iOn switch particularly well-suited for comparative lineage tracing in evo-devo contexts. Researchers can now ask how neural progenitor behavior differs across species or how

**Fig. 4 | Multicolor labeling using the iOn switch system across vertebrates. A** Turtle pallium electroporated at E14 and analyzed at E23 with six iOn construct variants (cytoplasmic and nuclear-targeted variants of *RFP*, *-YFP*, and *-TQ2*). The right images show an enlarged view of the boxed areas. **B** Rat cortex electroporated at E15 and analyzed at P7 with three-color iOn constructs (*iOnCAG ∞ TQ2*, *-EYFP*, and *-RFP*). The right images show an enlarged view of the boxed areas. **C** Guinea Pig cortex electroporated at E30 and analyzed at E52 with three-color iOn constructs (*iOnCAG ∞ TQ2*, *-EYFP*, and *-RFP*). The right images show an enlarged view of the boxed areas. **D** Zebrafish embryo injected at 0 dpf (one cell stage) and analyzed at 5 dpf with three-color iOn constructs (*iOnUbb ∞ TQ2*, *-EGFP*, and *-RFP*). The right images (**a**, **b**) show an enlarged view of the boxed areas.

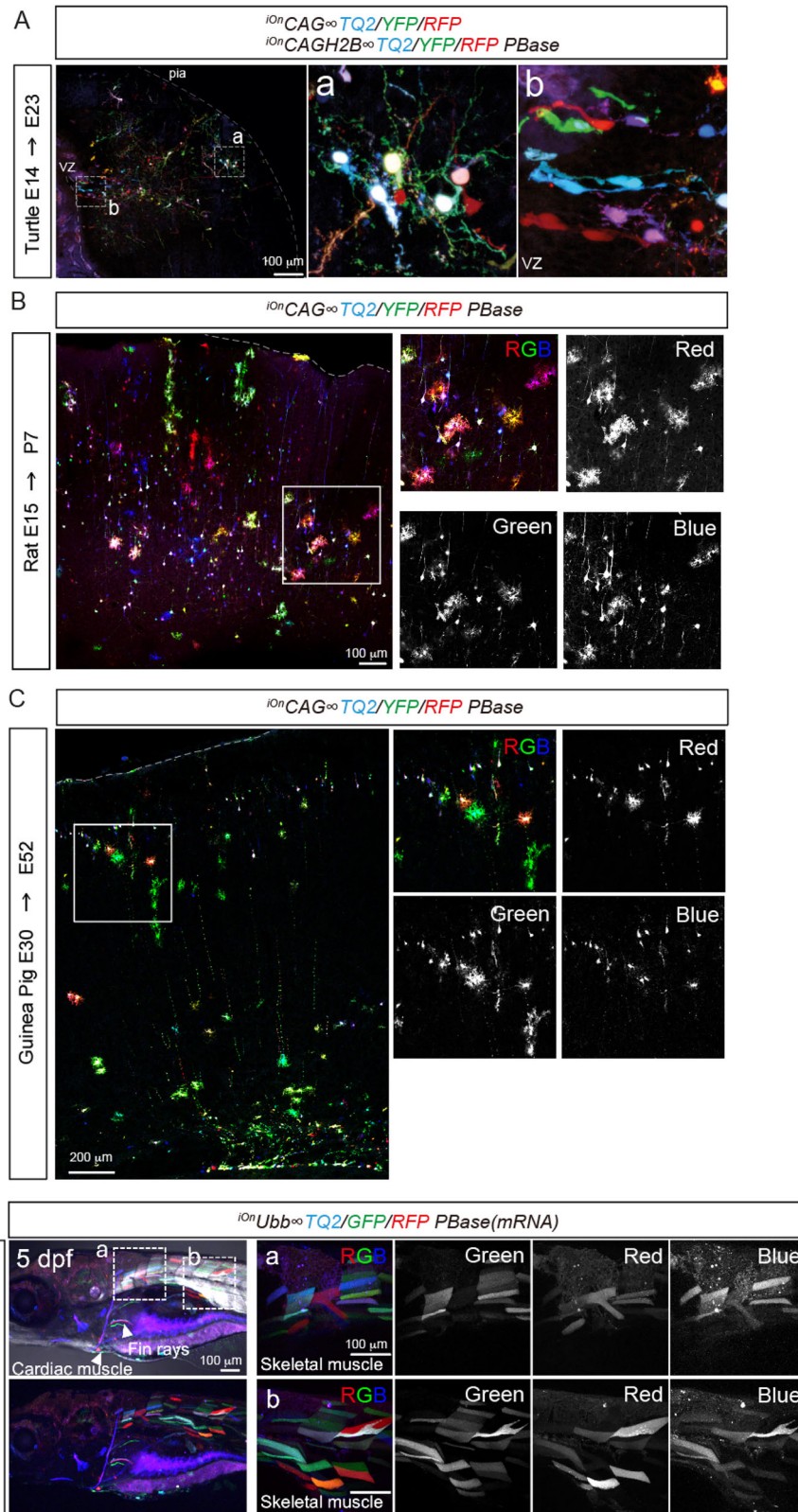

developmental strategies have evolved—without requiring species-specific transgenesis. To further support adoption of the iOn switch system across diverse model and non-model organisms, we provide a comprehensive summary of experimental parameters used throughout this study—including species, developmental stages, plasmid combinations, and electroporation settings—in Table S1. This resource is intended to serve as a practical reference for researchers seeking to implement the iOn system in their own lineage tracing experiments.

Because the iOn architecture is promoter-agnostic, it can be readily combined with inducible and cell-type–specific promoters. For example,

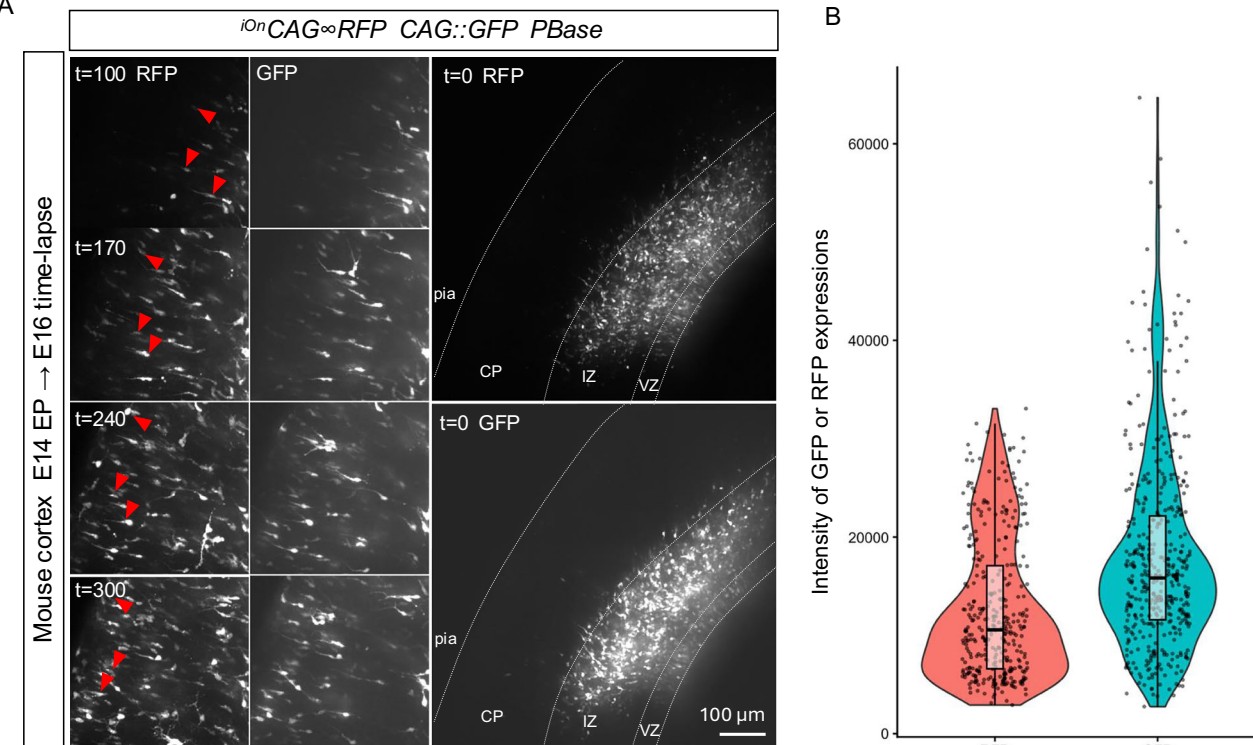

**Fig. 5 | iOn switch system supports uniform expression for live imaging. A** Mouse cortices were electroporated at E14 with both *ᶦᴼⁿCAG ∞ RFP* and *CAG::GFP*, dissected at E16, and subjected to time-lapse imaging over two days. Timepoints show representative RFP and GFP signals during neuronal migration. *CAG::GFP* expression is strong but highly variable, resulting in overexposure in cells electroporated and underexposure in other regions. In contrast, *ᶦᴼⁿCAG ∞ RFP* expression remains uniform, enabling consistent visualization across cortical areas and timepoints. **B** The violin plot compares expression distributions between the *CAG*-driven plasmid and the iOn switch, highlighting the advantage of integration-coupled activation for stable long-term imaging. Quantifications summarize embryo-level means from time-lapse imaging (mouse). *n* = 3 embryos. When per-cell trajectories are plotted, values were first averaged per movie, then per embryo, yielding one value per embryo. Imaging parameters. Objective (Olympus LUCPLFLN20x 20x, Dry 0.45), Interval: 600 [Sec] Iteration: 300 times.

iOn-TetON configurations would enable temporally controlled lineage tracing, and iOn cassettes driven by *Syn1* or *CaMKIIα* (neurons) or *hGFAP* or *Aldh1l1* (astrocytes) would permit selective labeling of defined cell populations. Intersectional variants such as iOn-CreERT2 could further combine cell-type specificity with ligand- or doxycycline-dependent timing, while preserving the key feature of iOn, integration-coupled activation with minimal episomal leak. Notably, iOn-driven Cre under the *Atoh7* promoter in the chick retina (*ᴸᶦᴼⁿAtoh7∞Cre*)[12] already provides proof-of-principle for cell-type-specific deployment.

The updated iOn switch system addresses longstanding bottlenecks in lineage tracing for developmental and comparative biology. By combining integration-dependent expression, plasmid-level modularity, tunable labeling density, multicolor compatibility, stable fluorescence for live imaging, and broad species accessibility, it provides a practical and scalable solution for modern Evo-Devo research. As developmental biology continues to diversify beyond traditional model systems, tools like iOn switch will be essential for enabling high-resolution, dynamic lineage analysis across the vertebrate phylogeny.

## Methods
### Animals
Mouse: Wild-type ICR mice were housed in the Waseda University Animal Facility, and all procedures were performed in strict accordance with institutional animal care guidelines. Mice were maintained under a 12-hour light/dark cycle with ad libitum access to food and water. The presence of a vaginal plug was designated as embryonic day (E) 0.5, and the day of birth as postnatal day (P) 0.

Chick: Fertilized chicken eggs were purchased from Yamagishi (Mie, Japan) and incubated at 38°C in a humidified incubator until the desired developmental stages were reached in Tokyo Metropolitan Institute of Medical Science.

Chinese softshell turtle (*P. sinensis*): Fertilized turtle eggs were purchased from a local breeder (Daiwa-youshoku, Saga, Japan) and incubated at 30°C in a humidified incubator until the desired developmental stages were reached in Tokyo Metropolitan Institute of Medical Science.

Rat: All animal procedures for rats were approved by the Institutional Safety Committee on Recombinant DNA Experiments and the Animal Research Committee of the University of Tokyo. WistarST rats were purchased from SLC Japan. Rats were housed in cages with bedding (Avidity Science, TEK-FRESH) and provided constant access to food (Nippon Crea, Rodent Diet CE-2) and water. The animal facility was maintained at a temperature of 23 ± 2°C, a humidity of 50 ± 10%, and a 12-hour light/dark cycle. Plug day was defined as embryonic day (E) 0.5, and the day of birth as postnatal day (P) 0. Data from all embryos were pooled without discrimination of sex, given the difficulty of determining sex identity at embryonic stages.

Guinea pig: Pregnant guinea pigs (Slc:Hartley) were purchased from SLC Japan. The day of vaginal plug detection was designated as embryonic day 0 (E0). All animal experiments for guinea pigs were carried out under the Guidelines for Laboratory Animals of Kumamoto University.

Zebrafish: Zebrafish (Danio rerio) were raised and maintained at the fish facility of the Bioscience and Biotechnology Center, Nagoya University, under a 14–10 h light-dark cycle at 28.5 °C. Wild-type zebrafish with the AB genetic background were used in this study. The zebrafish study was approved by the Nagoya University Animal Experiment Committee and was conducted in accordance with the Regulations on Animal Experiments at Nagoya University. We have complied with all relevant ethical regulations for animal use.

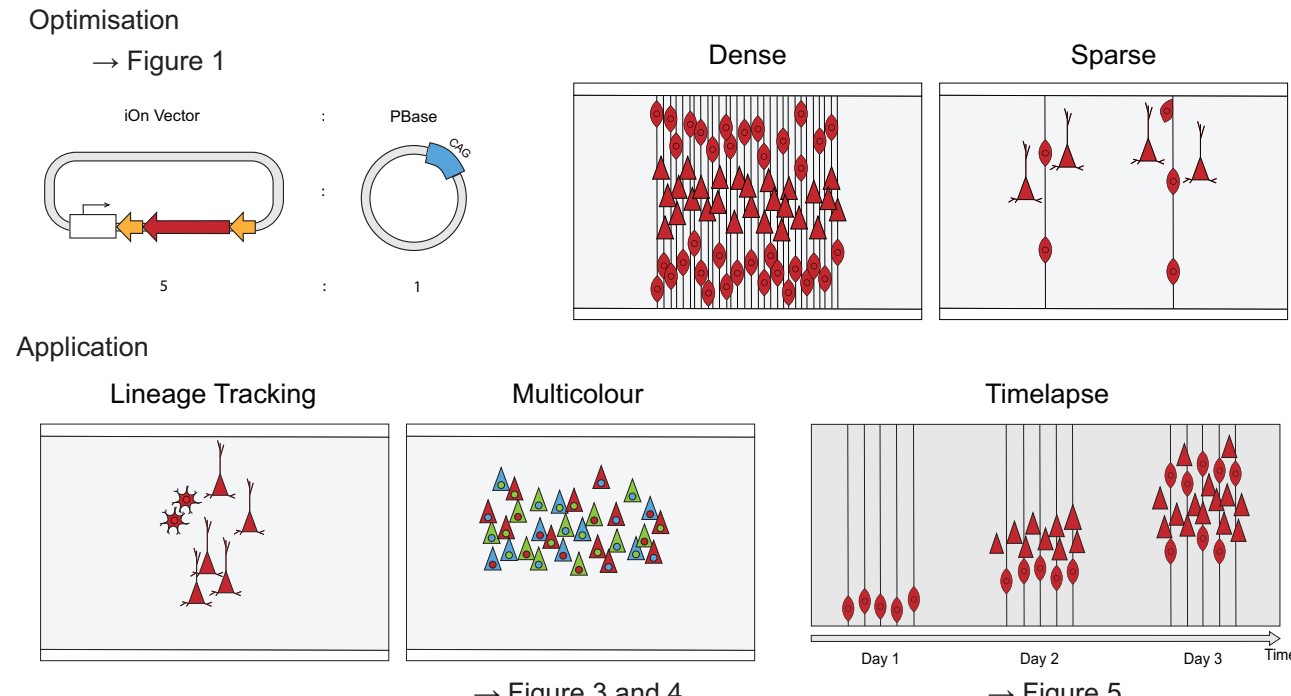

**Fig. 6 | Summary of improvements in the updated iOn switch system for lineage tracing and live imaging.** Schematic overview highlighting the key enhancements to the iOn switch system presented in this study. (Top panel): Optimization module illustrating (1) the ideal ratio of PiggyBac transposase (PBase) to iOn vector for efficient genomic integration with minimal cytotoxicity, and (2) the tunable control of labeling density by adjusting plasmid concentration—enabling both sparse single-clone labeling and dense multicolor reconstruction. (Bottom panel): Applications of the improved iOn switch system, including its use in (3) stable and combinatorial multicolor labeling for lineage reconstruction across species, (4) subcellular targeting for spatial resolution, and (5) consistent expression during long-term live imaging of dynamic developmental processes. Together, these advances establish the iOn switch toolkit as a flexible and accessible platform for evo-devo lineage tracing studies in both model and non-model organisms.

## Ethics statement

All animal experiments were conducted in compliance with relevant institutional guidelines and regulations. Mice: All procedures were approved by the Animal Care and Use Committee of the Tokyo Metropolitan Institute of Medical Science (TMIMS) and the Institutional Animal Care and Use Committee of Waseda University. Chick and turtle: All procedures were approved by the Animal Care and Use Committee of the Tokyo Metropolitan Institute of Medical Science (TMIMS). Rat: All procedures involving rats were approved by the Institutional Safety Committee on Recombinant DNA Experiments and the Animal Research Committee of the University of Tokyo. Guinea pig: All animal experiments involving guinea pigs were carried out following institutional guidelines and were approved by the licensing committee of Kumamoto University. Zebrafish: The zebrafish study was approved by the Nagoya University Animal Experiment Committee and was conducted in accordance with the Regulations on Animal Experiments at Nagoya University.

## Plasmid construction

To generate new fluorescent variants, each fluorescent protein (FP) sequence was PCR-amplified and cloned into the $^{iOn}CAG-MCS$ vector, as previously described[12]. FPs used in this study included EGFP (Clontech), mTurquoise2[31], EYFP[32], and MT-mKeima-Red[33]. For nuclear localization, the human histone H2B sequence[34] was fused in-frame to the 5' end of each FP coding sequence. To optimize the iOn plasmid for zebrafish, the *CAG* promoter was replaced with the zebrafish *ubiquitin B* (*Ubb*) promoter using the NEBuilder cloning strategy[35]. For *hyPBase* mRNA, capped RNA was synthesized using *pCS2+::hyPBase* plasmid as the template.

## Cell culture and transfection

HEK293 cells were obtained from ATCC and cultured in DMEM supplemented with 10% fetal bovine serum and 1% penicillin–streptomycin at 37 °C with 5% $CO_2$. Cells were routinely tested for mycoplasma contamination, and were negative. Cell line authentication by STR profiling was not performed for this study. HEK293 cells were maintained in Dulbecco's Modified Eagle Medium (DMEM; Gibco) supplemented with 10% fetal bovine serum (Hyclone), L-glutamine (Nacalai), and penicillin/streptomycin (Gibco), in a 5% $CO_2$ humidified incubator at 37 °C. For transfection, $1.5 \times 10^5$ cells were seeded onto collagen-coated coverslips in 24-well plates one day prior. Cells were transfected with iOn and PBase plasmids using 0.7 μL Lipofectamine 3000 (Invitrogen) per well. Medium was replaced four hours post-transfection. After 24–48 hours, cells were fixed with 4% paraformaldehyde (PFA) in phosphate-buffered saline (PBS), rinsed, and mounted in Fluoro-KEEPER Antifade Reagent (Nacalai Tesque, Japan). Images shown are representative of three or more independent experiments.

## Time-lapse imaging in cell culture

For live imaging, HEK293 cells were cultured on glass-bottom dishes. Embryonic brains were electroporated with *CAG::GFP* and $^{iOn}CAG \infty RFP$, dissected two days later, and embedded in 3% low-melting agarose in HEPES-buffered saline (HBS). The embedded brains were sectioned coronally at 300 μm using a LinearSlicer PRO7 (Dosaka). Slices were placed on membrane inserts (PICMORG50, Merck Millipore) and incubated in Neurobasal medium with B27 (Thermo Fisher Scientific) and antibiotics (Antibiotic-Antimycotic, Gibco) at 5% $CO_2$ and 60% $O_2$.

## Confocal time-lapse microscopy

Live imaging was performed using a Leica SP5 or SP8 inverted confocal microscope equipped with a 20× long working distance objective (HC PL FLUOTAR L 20×/0.40 CORR, Leica). Slices were cultured in a stage-top incubator (Chamlide TC; Live Cell Instrument or STXG-GSI2X; Tokai Hit). Z-stack images (10–15 planes at 10 μm intervals) were acquired at each time point, and maximum intensity projections were generated.

### In utero electroporation (mouse, rat, Guinea Pig)

Mouse: *In utero* electroporation was performed as described previously[12]. Pregnant ICR mice were anesthetized via intraperitoneal injection of a cocktail consisting of 5.0 mg/kg butorphanol (Meiji Seika), 4.0 mg/kg midazolam (Sandoz), and 0.75 mg/kg medetomidine (Nippon Zenyaku Kogyo). Uterine horns were exposed and kept moist with warm saline. Plasmid DNA mixed with 0.1% Fast Green (Wako) in HBS was injected into the lateral ventricles using pulled capillaries (GD-1, NARISHIGE). Afterward, the uterus was returned, and the incision was sutured. Dams were kept warm until recovery. For E18 samples, pregnant dams were euthanized by cervical dislocation, embryos were harvested, and embryonic brains were dissected after a brief wash in PBS. Brains were post-fixed in 4% PFA for 1 h. For cryosectioning, samples were sequentially immersed in 15% sucrose until submerged, followed by immersion in 30% sucrose in PBS at 4°C overnight. Brains were then embedded in Tissue-Tek O.C.T. compound (Sakura) and sectioned at 50 μm thickness using a cryostat. Sections were stored at –80 °C until imaging. For P7 samples, pups were anesthetized with isoflurane and transcardially perfused for fixation. Brains were then dissected, washed in PBS, post-fixed in 4% PFA for 1 h, and stored at 4 °C in PBS containing sodium azide (NaN₃) until use.

Rat: *In utero* electroporation was performed with modifications from established protocols[36]. Timed-pregnant Wistar ST rats (E15.5) were anesthetized with isoflurane. Plasmid solutions (1–1.5 mg/mL DNA) were injected into the embryonic lateral ventricles using heat-pulled capillaries. Electroporation was conducted using tweezer electrodes (Nepa Gene, Cat# CUY650P5) connected to a NEPA21 Type II electroporator (Nepa Gene) with the following settings: Voltage 50 V, pulse duration 50 ms, interval 950 ms, 4 pulses. Embryos were returned to the abdominal cavity, and the mothers were sutured and placed on a heating plate until recovery. For P7 samples, the dam was anesthetized with isoflurane, and the P7 pups were separated from her under anesthesia. P7 rat pups were anesthetized with isoflurane and perfused transcardially with ice-cold 4% PFA in PBS. Brains were then dissected and post-fixed in 4% PFA overnight at 4 °C. After fixation, samples were transferred to PBS and sectioned using a vibratome (Leica, Cat# VT1000S). The sections were stored in PBS at 4°C until further use.

Guinea pig: Pregnant guinea pigs (Slc:Hartley) used for *in utero* electroporation were purchased from SLC Japan. The day a vaginal plug was detected was designated as embryonic day 0 (E0). Timed-pregnant guinea pigs (14-15 weeks) were deeply anesthetized by intramuscular injection of a mixture of ketamine (46 mg/kg) and xylazine (24 mg/kg). Xylocaine jelly (2%, AstraZeneca) was used for local skin anesthesia. A DNA solution was introduced into the lateral ventricles of the embryos, and a square-wave electroporator (CUY21SC, NEPA GENE, Japan) was used to deliver four 50 ms pulses at 950 ms intervals at 70 V on E30[18]. After surgery, buprenorphine hydrochloride (8 μg/kg) was administered intramuscularly for postoperative analgesia. For E52 samples, animals were deeply anesthetized by intramuscular administration of the ketamine–xylazine mixture as described above and subsequently euthanized by inhalation of vaporized isoflurane until cardiac arrest was confirmed. Embryos were removed and euthanized by decapitation. Fetal brains were dissected and fixed with 4%PFA in PBS at room temperature for several hours and then incubated sequentially with 25% (w/v) sucrose in PBS (pH 7.4). After cryoprotection, brains were embedded in Tissue-Tek O.C.T. compound and frozen at –80 °C. 20 μm coronal sections were cut using a cryostat (Leica, CM1860).

For *in utero* electroporation, plasmids were prepared using endotoxin-free kits and verified for quality[27]. Per-plasmid concentration was kept at ~1 μg/μL, and total DNA mass was minimized when co-electroporating multiple constructs, in line with widely used IUE practices[25,28]. Electroporation parameters (pulse number, duration, and voltage) were optimized to achieve efficient transfection with minimal tissue stress[24,25]. Plasmids were prepared at 1 μg/μL in injection buffer, and 1 μL of the mixture was injected per embryo (total 1 μg DNA per embryo), except for the titration assay in Fig. 1.

### In Ovo Electroporation (Chick and Turtle)

Chick: *In ovo* electroporation was conducted following established protocols[37]. After the removal of albumen, the top of the E4 chick eggshell was opened. A DNA construct was directly injected into the ventricle with a mouth pipette. To inject DNA into the ventricle in pallium, electrodes were positioned at the top and bottom of the brain, and 15 V, 50 ms, 950 ms, 1 time was applied. Electric pulses were applied between electrodes using a BEX Gemini electroporator (BEX Co., Ltd.). The eggshell was sealed with tape and incubated until it reached the desired stage. For E6/E7 chick embryos, eggs were opened, and embryos were collected, washed in PBS, fixed in 4% PFA/PB for 1 h, and then dissected. Because E7 chick embryos are at an early developmental stage, euthanasia was not applicable. Brains were post-fixed in 2% PFA overnight. For cryosectioning, samples were sequentially immersed in 15% sucrose until submerged, followed by immersion in 30% sucrose in PBS at 4°C overnight. Brains were then embedded in Tissue-Tek O.C.T. compound and sectioned at 50 μm thickness using a cryostat. Sections were stored at –80 °C until imaging.

Turtle: *In ovo* electroporation was conducted following established protocols with slight modifications[17]. The top of the E14 turtle eggshell was opened. A DNA construct was directly injected into the ventricle with a mouth pipette. To inject DNA into the ventricle in pallium, electrodes were positioned at the top and bottom of the brain, and 15 V, 50 ms, 950 ms, 1 time was applied. Electric pulses were applied between electrodes using a BEX Gemini electroporator (BEX Co., Ltd.). The eggshell was sealed with parafilm and incubated until it reached the desired stage. Embryos were fixed with 4% PFA/PB. For E18/E23 turtle embryos, eggs were opened, and embryos were collected, washed in PBS, fixed in 4% PFA/PB for 1 h, and then dissected. Because E18/E23 turtle embryos are at an early developmental stage, euthanasia was not applicable. Brains were post-fixed in 2% PFA overnight. For cryosectioning, samples were sequentially immersed in 15% sucrose until submerged, followed by immersion in 30% sucrose in PBS at 4°C overnight. Brains were then embedded in Tissue-Tek O.C.T. compound and sectioned at 20 μm thickness using a cryostat. Sections were stored at –80 °C until imaging.

### Microinjection (Zebrafish)

To optimize the iOn plasmid for zebrafish, the *CAG* promoter was replaced with the zebrafish ubi promoter[35]. Three iOn switch plasmids (*$^{iOn}$pUbb::GFP, ::RFP, and ::TQ2*) were coinjected at 6.5 pg per plasmid into embryos at the one-cell stage, together with 30 pg of *PBase* mRNA. Injected larvae were screened live for GFP expression using a fluorescence stereomicroscope at 1 dpf or later.

### Tissue processing

Chick embryonic brains up to E7 were fixed by immersion in 4% PFA. For postnatal mouse brains, transcardial perfusion was performed using PBS followed by 4% PFA. Brains were post-fixed in 2% PFA overnight. For cryosectioning, samples were sequentially immersed in 15% sucrose until submerged, followed by immersion in 30% sucrose in PBS at 4°C overnight. Brains were then embedded in Tissue-Tek O.C.T. compound (Sakura) and sectioned at 50 μm thickness using a cryostat. Sections were stored at –80 °C until imaging.

### Image acquisition

Sections were rinsed in PBS and mounted using SlowFade™ Gold Antifade Mountant (Thermo Fisher Scientific). Imaging was performed using an LSM800 confocal microscope (Carl Zeiss). The excitation and emission settings for each fluorophore were as follows: mTurquoise2 (TQ2): Ex 458 nm/Em 480–510 nm; EYFP: Ex 514 nm/Em 525–555 nm; EGFP: Ex 488 nm/ Em 500–530 nm; RFP (mCherry): Ex 561 nm/Em 580–620 nm; iRFP: Ex 640 nm/Em 660–700 nm. Z-stacks were acquired at 4 μm intervals, and maximum intensity projections were generated using ZEN Blue software (Carl Zeiss). For zebrafish, a subset of GFP-positive animals was subsequently imaged live for GFP, RFP, and TQ2 expression using a Zeiss LSM700 confocal microscope. All image quantification and processing were

performed using ImageJ (NIH). For figure preparation, image levels were adjusted uniformly in Adobe Photoshop (Adobe Systems).

## Cell clustering and ternary plot visualization

Raw fluorescence intensity values of each cell were first obtained using Fiji (ImageJ) and subsequently normalized such that the sum of all of the fluorescence equaled 1. These compositional values were transformed using a centered log-ratio (CLR) to account for their constrained nature. Cluster analysis was performed using Gaussian mixture modelling implemented in the *mclust* R package and visualized on a ternary plot using the *ggtern* package. Wherever possible, convex hulls were computed in ternary space to enclose points belonging to the same cluster and overlaid as semi-transparent polygons. Each sample point was mapped to its original RGB ratio to visually reflect fluorescence composition.

## Statistics and reproducibility

Unless specified otherwise, n denotes biological replicates: animals (in vivo) or independent transfections performed on different days (cell culture). For in vivo assays, fields of view (FoVs) within sections were treated as technical repeats and averaged to yield a single value per embryo. For time-lapse imaging, multiple cells were summarized per movie, then per embryo. For cell-culture assays, multiple FoVs within a well were averaged to one value per transfection. Reagent ratios: Unless specified otherwise, reagent ratios are reported on a molar (pmol; copy-number) basis, with weight values in parentheses for reference. Moles of double-stranded DNA were computed as "pmol = mass(ng) × 1000 / (660 × length (bp))". Using this, $^{iOn}CAG \infty RFP$ (5,223 bp) at 1 μg equals 0.29 pmol, and *PBase* (6,846 bp) at 0.2 μg equals 0.044 pmol, i.e., PBase:iOn ≈ 1:6.59 (molar ratio). No statistical hypothesis testing was performed; data are presented descriptively and summarized as mean ± s.e.m. with individual data points. A priori exclusion criteria were predefined. Animals/experimental units were excluded only for technical reasons (e.g., unsuccessful electroporation/labeling, or insufficient imaging/segmentation quality). Exclusions occurred intermittently during data collection and were not systematically recorded as counts; however, all analyses report the final *n* values for biologically independent samples in the figure legends.

Confounders: To minimize batch and order effects, samples from different groups were processed in parallel where possible and distributed across experimental batches. Imaging and quantification were performed using identical acquisition and analysis settings for all groups.

## Reporting summary

Further information on research design is available in the Nature Portfolio Reporting Summary linked to this article.

## Data availability

Source data underlying the graphs and charts are provided as Supplementary Data 1 and 2. All other data supporting the findings of this study are available from the corresponding author upon reasonable request.

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

## Acknowledgements

We thank Dr. Leonard Zon for providing the plasmid pENTR5'_ubi (Addgene plasmid #27320). We thank Jean Livet and his team for their helpful comments on this project. This work was supported by a Leading Initiative for Excellent Young Researchers (LEADER) grant (grant number 2020L0019), JSPS KAKENHI grants (20K22665, 22H02638), the FY2021 Research Grant from the Takeda Science Foundation, and the FY2022 Research Grant from the Mochida Memorial Foundation for Medical and Pharmaceutical Research (to T.K.). K.W. was supported by JST SPRING (JPMJSP2121). Some schematic illustrations were created with BioRender.com.

## Author contributions

T.K. designed the research. Z-C.N., K.W., and T.K. performed cell culture, mice, chick, and turtle experiments. J.H. and H.S. performed experiments on guinea pig. Y.Y., P.R. and I.K.S. performed experiments for rat. T. Kaneko and M.H. performed experiments for zebrafish. Z-C.N. and T.K. wrote the article. C.H. and C.O-M. provided input through close discussions throughout the study and contributed to shaping the experimental design and interpretation. All authors reviewed and approved the final manuscript.

## Competing interests

The authors declare no competing interests.
