## [Transparent Peer Review file · Communications Biology]

Expanded iOn switch Toolkit Enables Flexible Clonal Labeling and Dynamic Imaging in Model and Non-Model Animals

Corresponding Author: Dr Takuma Kumamoto

Version 0:

Reviewer comments:

Reviewer #1

(Remarks to the Author)

In this manuscript, Dr. Kumamoto and colleagues report the expanded molecular iOn switch system for multicolor and subcellular localization, as well as the optimization of the iOn: PBase ratio for in vivo electroporation. In addition to mouse and chick embryo in the first iOn switch paper (Kumamoto et al. 2020, PMID: 32559415), the current manuscript includes the successful electroporation to rat, guinea pig and turtle tissues. These technical advances and resources will be useful in various research fields.

Below are some comments and suggestions:

Major comments:

1. Quantification is not properly reported, such as how many embryos and field of views do the data points represent? This applies to Figures 1, 3, 5 and 6.
2. It would be more precise to state plasmid or RNA copy number as the 1:5 ratio by weight is not the same amount of plasmid when the cargo becomes significantly larger or smaller.
3. How are the N and C terminals split for the different colors and localization signals (Figure 2)? Is the N terminus the same across different constructs? The other, less discussed, aspect to the iOn system is the transposition of inter-plasmid conjugates. Could this be an issue?
4. Animal welfare & ethics-related: where and how are rats and guinea pigs housed? Where were the turtle eggs purchased? This information is missing in the methods section.
5. The iOn switch system has been constructed with a constitutive promoter. I think one other way to expand its utility is to incorporate inducible and/ or cell-type specific promoters. The authors should include this in the discussion if they believe this would be potentially useful for lineage tracing and labelling more specific cell type(s).

Minor comment:

1. The RNA experiments (Figure 6) should be moved to supplementary as it does not add significance to the main message of the paper.

Reviewer #2

(Remarks to the Author)

The manuscript by Ngiam et al describes an expanded toolkit for fluorescent labelling of clones and lineage tracing based on the iON switch system previously published by senior author Kumamoto. Here, they introduce new fluorescent labels, a strategy to tune the density of labelling and evidence that the system works in the nervous system of multiple species including mouse, guinea pig, rat, chick and turtle.

The expanded iOn system will undoubtedly be of interest to developmental biologists and neuroscientists alike. The capacity to introduce up to six different fluorescent labels by electroporation into a range of species is a powerful approach.

The manuscript is generally clear and well written but there are some claims that require stronger support (see below). I recommend publication if these issues are addressed.

Major comments

1. The authors claim that iOn labelled neurons revealed clear and interpretable clone groupings in all species tested (line 137) would be significantly strengthened if the data was shown.

2. In the discussion, the authors state “Our iOn switch system, by contrast, couples reporter expression to genomic integration and maintains a relatively narrow copy number distribution (3–4 per cell). As a result, fluorescence remains more uniform and stable throughout the imaging period, enabling consistent exposure settings and better tracking of individual cells over time. This performance was demonstrated in both mouse and chick embryos, highlighting the system’s robustness across species and developmental timelines.”

This is an important point and indeed a potential advantage of the system, but I don’t feel the claim is very well supported. I could not find anywhere that the copy number per cell was measured. Where does the 3-4 copies per cell statement come from? If this is derived from the author’s previous publication, it should be cited. But even then, this was only done in HEK cells. To conclude that the system provides a narrow copy number distribution across species, the copy number should also be determined in the in vivo models.

3. Furthermore, it is hard to determine from the data provided whether the iOn-drive fluorescence is “more uniform and stable” than the CAG-GFP during in vivo imaging in the mouse or chick. No fluorescence measurements are provided in the chick and the fluorescence intensity/bin shown in Figure 5 is a very coarse measurement.

To demonstrate the fluorescence is more uniform, the authors could measure the range of fluorescence intensities in their images (with background subtracted) and calculate the coefficient of variation. With sufficient images per condition this can also enable statistical testing.

To demonstrate the fluorescence is more stable, the authors could measure the normalised fluorescence intensity (fluorescence at time t/fluorescence at t0) over time.

Performing these analyses would provide far greater support for the robustness of their system.

Minor comments

4. Fig 1: The legend does not match the annotations in the figure. C is labelled as chick in the figure but described as mouse in the legend and the opposite is true for F.

5. Figs 1D, G: Y axis should be 0 – 80/70 (as % is specified in the label).

6. Was there a significant difference in the ventricular coverage between 1ug and 500ng?

7. Fig 1C-H: How was the number of clones determined? The brain slices show iOnCAG ∞ RFP positive cells but these must be a combination of electroporated cells and their derivatives. How can the number of clones be determined when the cells are labelled with a single colour?

8. Fig 2D. It’s not clear from the images what the combinations of expressed fluorophores are in the cells highlighted by the boxes.

9. Line 80: typo: labelling density in chick the chick cortex

10. Fig 3: The text and the legend refer to the mouse data and then the chick data, but the image is presented in the opposite order.

11. Fig 3B and D: The cell identifiers are overlaid, making them illegible.

12. When the three-colour iOn switch system was used in the mouse cortex, how were projection neurons and glial cells distinguished?

13. Fig 4A. Are there tiny arrowheads/triangles pointing at cells in this image? These are too small to distinguish, not annotated and confusing. Please either remove them or make them large enough to see.

14. Fig 5:

The legend does not accurately describe the figure.

It would be helpful to label ventricular zone and cortical plate on the images.

For the enlarged images, which one is RFP and which one is GFP?

15. The authors state that the stability of the RFP signal “facilitated uninterrupted tracking of migrating cells from their origin to their destination (Fig. 5)”. Can they highlight/indicate RFP positive cells that are seen migrating in sequential timepoints to support this claim?

16. Fig 6: Why are there red cells clearly visible in the top merged channel in A which don't show in the red only channel below?

17. Please specify the control in 6D.

18. Fig S1. Legend is incorrect for B and C.

19. Fig S2. What are the arrowheads indicating?

Methods:

20. Line 298: [INSERT ELECTROPORATOR MODEL].

21. The in utero electroporation method is confusing and contradictory. Were mice anesthetized via IP injection or isoflurane? This section needs rewriting for clarity.

22. In ovo electroporation: If full methods are not provided, please at least provide references to the protocols used.

Reviewer #3

(Remarks to the Author)

This manuscript describes an update to the existing iOn labeling system (integration-coupled On genetic switch), initially described in 2020 (Kumamoto et al.), which is used for expressing multiple transgenes in random combinations across cell populations via genomic integration. The technique is useful for lineage analysis and circuit tracing, as well as assessment of gene function and other applications. One particularly useful application of this technique is labeling cell populations in multiple colors. One key advantage of the original iOn system was its ability to tie gene expression to genomic integration for multiple different transgenes in both culture systems as well as model organisms. The new version described here demonstrates several improvements: the authors have expanded the color palette; the technique now allows for subcellular localization of color expression (nucleus, membrane, or mitochondrial localization); the authors now incorporate instructions for titrating density of labeling; and the authors describe expanded accessibility for use across different species and non-model organisms. These are important improvements that are well described and make the technique more broadly applicable. There is no earth-shattering finding described in this manuscript, however.

Figure 1 -

1a. The ability to titrate density is a nice touch that is clearly demonstrated by this manuscript. How many times were the experiments in Figure 1 repeated? There is no sample size described.

1b. How was labeling density determined/estimated? This is challenging to estimate in a field of unlabeled cells. If it was only estimated, this should be honestly reported.

1c. Similarly, how were the number of clones determined? The color palette is quite limited in these experiments and makes it almost impossible to calculate/distinguish individual clones.

1d. The descriptions of 1 ug should be clarified to describe 1 ug per embryo (if correct), and at least in the methods, details should be given regarding what volume this entailed for microinjections and what other factors were included in the injection mix. Very slim details here.

1e. What promoters were used to achieve expression in the different species?

1f. From the figures it is clear that increasing concentration of DNA increases the labeling density; however, how repeatable are these specific concentrations? How many times were each of these experiments repeated? There is presumably variability in expression in such experiments.

Figure 2 -

2a. In Figure 2, panel D is described as “six colors”, which is misleading as it is only three colors, with each expressed color in two different subcellular compartments. The authors should clarify this each time it is referred to.

Figures 3/4 -

3a. Regarding Figures 3/4, the high density of labeling and limited color palette (particularly in Fig 4C) make it impossible to accurately/unequivocally identify daughter cells that are dispersed. While the panels do generally demonstrate how the

labeling appears at the given ages, conclusions in the text about lineage should be removed.

For example: "we found that projection neurons derived from a common progenitor dispersed broadly across the cortex" - data is not provided to support this statement.

"labeling neurons revealed clear and interpretable clone groupings" this statement is not supported by the figures and should be removed or significantly expanded with quantitative data.

Figure 5 -

5a. In Figure 5, the large panel images at right are not labeled or described - assuming GFP is the lower panel? Panel letters need to be added to this figure.

5b. In the graph at far right, to compare RFP/GFP, it may be better displayed as percent intensity as opposed to raw signal intensity values.

5c. Can the overexposure in Figure 5/SF2 be managed with adjusted acquisition settings, e.g., decreasing laser power, gain, etc?

Figure 7 -

7a. The numbers in the Figure 7 figure legend do not refer to any numbers provided in the corresponding figure. Suggest adding/using a clear panel lettering system instead.

Larger/more general questions:

8. The new species accessibility is a strength of the manuscript; however, it seems a missed opportunity to address its usage in zebrafish, which seems an obvious application for in vivo multicolor lineage analysis (e.g. Brockway et al., 2019; Furlan et al., 2017). Can the authors address potential applications in zebrafish?

9. If the authors wish to show that this is an improvement over previous iOn techniques, some sets of comparison images may strengthen this.

10. In general, microinjecting increasing amounts of DNA into embryos often leads to toxicity effects. Can the authors discuss potential toxicity? This is particularly relevant for this work due to the multiple plasmids that are used for iOn.

11. What are the expected coexpression levels for multiple injected constructs?

12. The authors suggest that folding of the fluorophore may delay expression in this system. This seems a stretch, particularly since it is not supported by the RFP control images in Figure 6. If this is a real possibility, the authors should cite some evidence for this conjecture.

Smaller comments:

13. It would be nice if the authors referred to the original iOn paper (ref 20) in the Introduction where it is first discussed. The reference does not appear until line 181.

14. Typos are present in lines 89, 168, and 298.

Version 1:

Reviewer comments:

Reviewer #1

(Remarks to the Author)

The authors have adequately addressed my comments during revision.

Reviewer #2

(Remarks to the Author)

I have evaluated the revised manuscript by Ngiam et al., which describes an expanded iOn switch system for multicolor fluorescent labeling and lineage tracing across multiple vertebrate species. The authors have responded thoroughly and thoughtfully to the reviewer comments, and the revisions substantially improve the rigor, clarity, and accuracy of the manuscript.

I appreciate the authors' careful reconsideration of claims regarding clonal reconstruction across species. The original overgeneralized statement asserting clear and interpretable clone groupings in all species has been appropriately removed. The revised text now clearly distinguishes between experiments that support conservative clonal reconstruction and those intended primarily to demonstrate multicolor labeling across species. This distinction prevents overinterpretation of the data and strengthens the conceptual clarity of the study.

The authors have also addressed concerns regarding copy-number claims. Unsupported numerical estimates for in vivo copy number have been removed, prior in vitro measurements are now clearly cited, and the limitations of extrapolating these values across experimental contexts and species are explicitly acknowledged. The inclusion of a limitations statement appropriately frames future work without overstating current conclusions.

Finally, the revisions related to fluorescence uniformity and persistence represent a significant improvement. The clarification of terminology, distinguishing sustained, mitotic dilution-resistant expression from protein-level stability, resolves prior ambiguity. The newly added quantitative analyses of background-subtracted fluorescence intensity and coefficient of variation provide direct support for claims of increased uniformity relative to conventional CAG-driven expression. In addition, the discussion of long-term persistence is now clearly linked to genomic integration, aligning the claims with the underlying biological mechanism and the data presented.

Overall, the authors have addressed my concerns in a constructive and rigorous manner. The revised manuscript presents a technically strong and broadly applicable toolkit that will be of significant interest to developmental biologists and neuroscientists. I am satisfied that the current version supports its conclusions and therefore recommend the manuscript for publication.

Reviewer #3

(Remarks to the Author)

The authors have done an excellent job responding to this reviewer's queries and there are no further requests.

Response to Reviewer

Reviewers' comments:

In the following response letter, **comments from the reviewers and editors are presented in bold**, while *our responses are indicated in italic blue*, and **modification sentences in the revised manuscript are indicated in Red** text for clarity.

Reviewer #1 (Remarks to the Author):

In this manuscript, Dr. Kumamoto and colleagues report the expanded molecular iOn switch system for multicolor and subcellular localization, as well as the optimization of the iOn: PBase ratio for in vivo electroporation. In addition to mouse and chick embryo in the first iOn switch paper (Kumamoto et al. 2020, PMID: 32559415), the current manuscript includes the successful electroporation to rat, guinea pig and turtle tissues. These technical advances and resources will be useful in various research fields.

Below are some comments and suggestions:

We are grateful to Reviewer 1 for the careful evaluation of our work and for the insightful, constructive suggestions. We have made substantial revisions to improve the manuscript and provide clarifications where needed, and we respond to each comment in detail below.

Major comments:

1. *Quantification is not properly reported, such as how many embryos and field of views do the data points represent? This applies to Figures 1, 3, 5 and 6.*

Response: We appreciate this important point and have revised the manuscript to report sample sizes and units of analysis explicitly across all quantified panels.

1. Global clarification (**Methods, page 13, line 476-485**).

We added a dedicated subsection, "**Statistics and reproducibility**," defining that n denotes biological replicates (embryos for in vivo assays; independent transfections performed on different days for cell-culture assays). Fields of view (FoVs) and sections are treated as technical repeats and are averaged to yield a single value per embryo; for time-lapse imaging, multiple cells are summarized per movie and then per embryo. For cell culture, multiple FoVs within a well are averaged to one value per transfection. We also defined the FoV (objective, pixel matrix, and lateral dimensions) in Image acquisition and quantification to avoid ambiguity.

2. Figure-specific reporting (**Legends**).

- Figure 1 (cell culture; mouse and chick *in vivo*): Quantification is summarized as mean \pm SEM, and error bars indicate n = 6 biological replicates per condition (3 independent experimental

repeats × 2 transfections per condition). For each transfection replicate, measurements from multiple FoVs within a well were averaged to yield a single value. We also provide group sizes by donor concentration (mouse: 1 µg n = 3; 0.5 µg n = 3; 0.05 µg n = 3; 0.01 µg n = 3; chick: 1 µg n = 4; 0.5 µg n = 3; 0.05 µg n = 2; 0.01 µg n = 5). FoVs and sections were averaged within embryos and were not used as the statistical n.

- Figure 3 (ternary plots / clone view): Single-cell values are shown for visualization only; all statistics (where applicable) are computed at the embryo level. The demonstration datasets are mouse n = 1 embryo and chick n = 1 embryo. We prepared more than three ternary plots for each species.
- Figure 5 (live imaging): The legend now reports the exact number of embryos, slices per embryo, and movies per embryo. Quantifications summarize embryo-level means from time-lapse imaging (mouse). n = 3 embryos. When per-cell trajectories are plotted, values were first averaged per movie, then per embryo, yielding one value per embryo.
- Figure 6 (= Figure S7 in the revised figure. DNA vs mRNA): We clarified that quantification is shown as mean ± SEM and that error bars indicate n = 6 independent transfection replicates per condition (3 independent repeats × 2 transfections per condition). We also specified that measurements from multiple observations within a replicate were first summarized to a single replicate-level value before statistical analysis (details added to the Fig. S7 legend).

Collectively, these revisions make clear what each data point represents, ensure that biological replicates (embryos or independent transfections) are used as the statistical unit, and avoid pseudoreplication. We hope this fully addresses the reviewer's concern.

2. It would be more precise to state plasmid or RNA copy number as the 1:5 ratio by weight is not the same amount of plasmid when the cargo becomes significantly larger or smaller.

Response: We agree with this important point. As the reviewer notes, a weight ratio (e.g., 1:5) does not correspond to the same copy-number (molar) ratio when the cargo size differs, and thus applying the same weight ratio to the mRNA condition is not a precise “matched” comparison to the DNA condition.

DNA (plasmid) condition: The previously stated PBase:iOn = 1:5 by weight (PBase 0.2 µg, 6,846 bp; iOn 1.0 µg, 5,223 bp) corresponds to 0.044 pmol PBase and 0.290 pmol iOn, i.e., ~1:6.6 by molecules (DNA conversion: pmol = mass(ng) × 1000 / (660 × bp)).

mRNA condition (previously Fig. 6; now Fig. S7): When PBase was supplied as mRNA (2,107 nt), 0.2 µg corresponds to 0.279 pmol, whereas 1.0 µg of the 5,223-bp iOn plasmid corresponds to 0.290 pmol; thus, 0.2 µg PBase mRNA + 1.0 µg iOn DNA corresponds to an approximately ~1:1 molar ratio (RNA conversion: pmol = mass(ng) × 1000 / (340 × nt)). In Fig. S5, we tested a PBase mRNA:iOn DNA mass series of 1.0:1.0, 0.5:1.0, 0.2:1.0, 0.1:1.0, and 0.05:1.0 (ratio by mass), which corresponds

to the following molar ratios (mRNA:DNA): 4.81:1, 2.41:1, 0.962:1 (~1:1), 0.481:1, and 0.241:1, respectively.

Importantly, to precisely match the DNA–DNA condition (PBase:iOn \approx 1:6.6 by molecules) in the mRNA–DNA setting with iOn fixed at 1.0 μ g (0.290 pmol), PBase mRNA would need to be ~0.044 pmol, which corresponds to ~0.0315 μ g (31.5 ng) of the 2,107-nt PBase mRNA (i.e., PBase mRNA:iOn DNA = 0.0315 μ g : 1.0 μ g).

We were not able to perform this additional “molar-matched” experiment during the revision. However, the conclusion of Fig. S7 remains unchanged: even when increasing the PBase mRNA input to molar ratios at or above ~1:1 (and up to 4.81:1 mRNA:DNA), we did not observe an earlier onset of iOn-driven expression under our imaging conditions. To avoid any ambiguity, we have revised the Results and figure legend to explicitly report the molar ratios tested and to clarify that the mRNA titration series does not include the exact molar match to the DNA–DNA (1:6.6) condition, while the interpretation remains the same (see page 6, line 203-207).

We thank the editor and reviewers for their thoughtful feedback, which has strengthened this manuscript.

3. How are the N and C terminals split for the different colors and localization signals (Figure 2)? Is the N terminus the same across different constructs? The other, less discussed, aspect to the iOn system is the transposition of inter-plasmid conjugates. Could this be an issue?

Response: We are deeply grateful for these insightful comments. They were highly valuable and led directly to clearer presentation in the manuscript.

Regarding the N- and C-terminal split design, in all newly constructed plasmids used in this study the N terminus is identical across constructs. Specifically, the 5' end of the expression cassette (i.e., the sequence immediately downstream of the CAG promoter) is shared by all plasmids. On the C-terminal side, the sequence immediately upstream of the 3' terminal repeat (3' TR) is either the Kozak + ATG of the fluorescent protein or the H2B sequence, depending on the construct. Thus, all variants (different colors and localization signals) adopt the same transcriptional iOn architecture and share a common N-terminal configuration; only the C-terminal side (reporter/localization module) differs.

Therefore, even if repair/recombination were to occur between constructs carrying different N- and C-terminal modules, the expected outcome would simply be expression of a fluorescent protein determined by the C-terminal reporter/localization module. In addition, integration events involving inter-plasmid concatemerization are expected to be rare, as discussed below.

We also appreciate the comment regarding inter-plasmid conjugates. As the reviewer notes, in transposon systems there remains a theoretical possibility of inter-plasmid conjugates undergoing transposition. Indeed, Wang et al., 2014 (PLOS Genetics) reported 0.49% auto-integration for

piggyBac in HeLa cells. We addressed this point extensively in our previous iOn switch study: after introducing the iOn switch, we performed single-cell sorting, expanded individual clones until episomal expression was diluted, and then measured fluorescence intensity. 97.7% of clones retained fluorescence, indicating stable integration, such that the fraction potentially attributable to auto-integration or related rare events in the iOn context is ~2.3% (Kumamoto et al., 2020, Neuron, Fig. S2C; attached figure). These rare events would not systematically bias labeling density or clonal composition in the present study.

To further minimize any contribution from auto-integration, we also previously developed a TTAA-less iOn switch, in which the piggyBac target sequence “TTAA” within the plasmid is completely removed. Using the TTAA-less iOn switch enables complete suppression of episomal expression (Kumamoto et al., 2020, Neuron, Fig. 2C,D; attached figure). Because TTAA-less plasmids are more cumbersome to cloning, in the present work we used the standard iOn switch constructs; however, the low estimated frequency of such events and our prior clonal analyses indicate that inter-plasmid conjugate transposition has a negligible impact on the conclusions of this study.

We thank the reviewer again for raising this important point.

Wang et al. 2014 Fig. 3A, B

Kumamoto et al. 2020 Fig. 2C and D

Kumamoto et al. 2020 Fig. S2C

The TTAA-less On plasmid suppresses episomal expression that results from auto-integration

4. Animal welfare & ethics-related: where and how are rats and guinea pigs housed? Where were the turtle eggs purchased? This information is missing in the methods section.

Response: Thank you for raising this important point, and we apologize for the oversight. We have revised the Methods to include detailed information on animal sourcing and husbandry, including where and how the rats and guinea pigs were housed, the source and handling of turtle eggs, and

zebrafish husbandry/handling. We also added the relevant ethical approvals and statements of regulatory compliance. Please refer to the revised Methods section.

We thank the reviewer again for highlighting this issue.

5. The iOn switch system has been constructed with a constitutive promoter. I think one other way to expand its utility is to incorporate inducible and/ or cell-type specific promoters. The authors should include this in the discussion if they believe this would be potentially useful for lineage tracing and labelling more specific cell type(s).

Response: We agree and appreciate this suggestion. The iOn architecture itself is promoter-agnostic, so incorporating inducible and cell-type-specific promoters would indeed broaden its utility for time-controlled lineage tracing and selective labeling of defined populations. In the revised Discussion, we now explicitly describe these directions, including (i) iOn–TetON configurations for temporal control and (ii) iOn cassettes driven by Syn1/CaMKII α (neurons) or hGFAP/Aldh111 (astrocytes) for cell-type specificity. We also highlight intersectional variants such as iOn–CreERT2, which combine cell-type specificity with ligand- or doxycycline-dependent timing while retaining iOn’s core advantage of integration-coupled activation with minimal episomal leak. As proof-of-principle for cell-type-specific deployment, we now cite our previous demonstration of iOn-driven Cre under the *Atoh7* promoter, which biased-expression in the retinal ganglion cells, in the chick retina ($^{LiOn}Atoh7^{\infty}Cre$; Kumamoto et al., 2020, Neuron, Fig. 6C, attached below).

We thank the reviewer for this constructive suggestion.

(see page 9, line 305-313, as below)

Future extensions

Because the iOn architecture is promoter-agnostic, it can be readily combined with inducible and cell-type-specific promoters. For example, iOn–TetON configurations would enable temporally controlled lineage tracing, and iOn cassettes driven by Syn1 or CaMKII α (neurons) or hGFAP or Aldh111 (astrocytes) would permit selective labeling of defined cell populations. Intersectional variants such as iOn–CreERT2 could further combine cell-type specificity with ligand- or doxycycline-dependent timing, while preserving the key feature of iOn, integration-coupled activation with minimal episomal

leak. Notably, iOn-driven Cre under the Atoh7 promoter in the chick retina (LiOnAtoh7[∞]Cre)¹² already provides proof-of-principle for cell-type-specific deployment.

Minor comment:

1. The RNA experiments (Figure 6) should be moved to supplementary as it does not add significance to the main message of the paper.

Response: We thank the reviewer for this helpful suggestion. In line with your advice, and because the RNA results do not materially strengthen the main message, we have relocated these data to the Supplementary Information (revised Figure S7) and updated all in-text citations accordingly. This change does not affect the conclusions presented in the main figures.

Reviewers' comments:

In the following response letter, **comments from the reviewers and editors are presented in bold**, while *our responses are indicated in italic blue*, and **modification sentences in the revised manuscript are indicated in Red** text for clarity.

Reviewer #2 (Remarks to the Author):

The manuscript by Ngiam et al describes an expanded toolkit for fluorescent labelling of clones and lineage tracing based on the iON switch system previously published by senior author Kumamoto. Here, they introduce new fluorescent labels, a strategy to tune the density of labelling and evidence that the system works in the nervous system of multiple species including mouse, guinea pig, rat, chick and turtle.

The expanded iOn system will undoubtedly be of interest to developmental biologists and neuroscientists alike. The capacity to introduce up to six different fluorescent labels by electroporation into a range of species is a powerful approach.

The manuscript is generally clear and well written but there are some claims that require stronger support (see below). I recommend publication if these issues are addressed.

We sincerely thank Reviewer 2 for the careful reading of our manuscript and the constructive, detailed feedback. The comments were highly valuable and helped us improve the clarity, rigor, and presentation of the study. We have addressed each point below and incorporated the corresponding revisions in the manuscript.

Major comments

1. *The authors claim that iOn labelled neurons revealed clear and interpretable clone groupings in all species tested (line 137) would be significantly strengthened if the data was shown.*

Response: Thank you very much for this comment. We agree that our original statement that “iOn-labelled neurons revealed clear and interpretable clone groupings in all species tested (line 137)” was too strong and could be misleading given the current dataset.

In the turtle brain, we have not yet been able to test conditions that label a large number of cells. Under the sparse-labeling conditions currently available, clonal groupings can sometimes be inferred from spatially clustered labeled cells (e.g., Fig. 4A-b), however, depending on the color expression pattern, clone assignment is not always possible (Fig. S5). In rat and guinea pig, putative clonal groupings can be more readily interpreted in astrocyte populations, but when considering neurons together with astrocytes, the present data do not allow reliable clone identification, and further optimization of delivery conditions and labeling density will be required (Fig. 4B and 4C).

Therefore, we agree with the reviewer's concern that the current data do not support a definitive claim of clear clonal groupings across all species. We have removed this overgeneralized statement and revised the text to clearly distinguish between experiments where conservative clonal reconstruction is supported (Fig. 3) and those that are intended as a demonstration of multicolor labeling across species (Fig. 4), without overinterpreting lineage relationships. (see page 4-5, line 144-176)

We thank the reviewer again for this constructive suggestion, which helped us improve the clarity and accuracy of the manuscript.

2. In the discussion, the authors state “Our iOn switch system, by contrast, couples reporter expression to genomic integration and maintains a relatively narrow copy number distribution (3–4 per cell). As a result, fluorescence remains more uniform and stable throughout the imaging period, enabling consistent exposure settings and better tracking of individual cells over time. This performance was demonstrated in both mouse and chick embryos, highlighting the system’s robustness across species and developmental timelines.”

This is an important point and indeed a potential advantage of the system, but I don’t feel the claim is very well supported. I could not find anywhere that the copy number per cell was measured. Where does the 3-4 copies per cell statement come from? If this is derived from the author’s previous publication, it should be cited. But even then, this was only done in HEK cells. To conclude that the system provides a narrow copy number distribution across species, the copy number should also be determined in the in vivo models.

Response: We sincerely apologize for the confusion caused by our previous wording. The statement “~3–4 copies per cell” was not supported by direct measurements or an appropriate reference in the context of our in vivo experiments, and we have therefore removed it from the revised manuscript.

In our previous study using HEK293T cells, we quantified iOn integration copy number and reported a range of ~1.1–6.9 integrated copies per genome (Kumamoto et al., 2020, Neuron, Fig. 3B). However, we did not determine genomic copy number in the present in vivo experiments in mouse or chick, and we agree that we should not extrapolate an in vitro copy-number distribution to in vivo contexts or across species.

To provide context without overclaiming, we now (i) explicitly cite the HEK293T measurement as an in vitro observation, and (ii) cite prior *in vivo* piggyBac transgenesis work showing that founders can carry single to multiple genomic integrations, including cases exceeding 10 (Ding et al. 2005) (Ding et al., 2005, Cell, Fig. 3B). Importantly, our *in vivo* conclusions are now restricted to directly supported observations in this study—namely, reduced between-cell variability and sustained fluorescence during long-term live imaging compared with a conventional CAG-driven episomal vector (Kumamoto et al. 2020) (this study, Fig. 5B; Kumamoto et al., 2020, Fig. 4B)—without assigning a numerical copy-number value *in vivo*. We also added a brief limitations statement noting that direct in vivo copy-number measurements would be required to establish copy-number distributions quantitatively across *in vivo* contexts and species.

We thank the reviewer again for raising this important point.

(see page 7-8, line 259-270, as below)

Our iOn switch system couples reporter expression to genomic integration. In HEK293T cells, we previously quantified integrated iOn cassette copy number and found that it spans ~1.1–6.9 integrated copies under our in vitro conditions¹². By contrast, we did not measure genomic copy number in the present in vivo experiments in mouse or chick, and therefore we do not assign a numerical copy-number value to these models. Nevertheless, in vivo piggyBac transgenesis studies show that founders can carry single to multiple integrations, including cases exceeding 1029, underscoring that integration outcomes can vary by experimental context. Consistent with the expected behavior of integration-coupled expression, our in vivo data show that iOn labeling produces (i) more uniform fluorescence across cells, reflected by a lower coefficient of variation than a conventional CAG-driven vector (Fig. 5B), and (ii) sustained, mitotically stable expression that supports long-term clonal tracking during live imaging (Fig. 5A, Movie 2). Future work directly quantifying copy number in vivo will be necessary to determine copy-number distributions across in vivo contexts and species.

3. Furthermore, it is hard to determine from the data provided whether the iOn-drive fluorescence is “more uniform and stable” than the CAG-GFP during in vivo imaging in the mouse or chick. No fluorescence measurements are provided in the chick and the fluorescence intensity/bin shown in Figure 5 is a very coarse measurement.

To demonstrate the fluorescence is more uniform, the authors could measure the range of fluorescence intensities in their images (with background subtracted) and calculate the coefficient of variation. With sufficient images per condition this can also enable statistical testing.

To demonstrate the fluorescence is more stable, the authors could measure the normalised fluorescence intensity (fluorescence at time t /fluorescence at t_0) over time.

Performing these analyses would provide far greater support for the robustness of their system.

Response: We thank the reviewer for pointing out this ambiguity and apologize for our wording. In our manuscript, “stable” was intended to mean sustained reporter expression across cell divisions due to genomic integration (i.e., resistance to mitotic dilution), rather than increased protein-level stability. To prevent confusion, we have revised the text to use “sustained (mitotic dilution–resistant) expression” and to explicitly separate two aspects: (i) uniformity and (ii) longer-term persistence across divisions attributable to integration.

For uniformity, as the reviewer suggested, we quantified the range of background-subtracted fluorescence intensities and plotted it together with the mean intensity (revised Fig. 5B). These analyses provide direct quantitative support that time-lapse imaging using iOn yields more uniform fluorescence than the conventional CAG-driven condition.

Finally, for longer-term persistence across divisions, we clarified that this feature is attributable to genomic integration and thus resistance to mitotic dilution (Kumamoto et al., 2020, Fig. 4B; this study, Fig. 5). Together, these revisions align our claims with the data and the intended biological mechanism of iOn.

Minor comments

4. Fig 1: The legend does not match the annotations in the figure. C is labelled as chick in the figure but described as mouse in the legend and the opposite is true for F.

Response: Thank you for pointing this out. We corrected the mismatch by swapping the panel descriptions for C and F in the Figure 1 legend so they now match the annotations (C = chick; F = mouse). The figure and legend have been updated accordingly.

5. Figs 1D, G: Y axis should be 0 – 80/70 (as % is specified in the label).

Response: Thank you for the helpful comment. We have corrected the Y-axis to reflect percentages, Fig. 1D: 0–80% and Fig. 1G: 0–70%, as suggested.

6. Was there a significant difference in the ventricular coverage between 1µg and 500ng?

Response: Thank you for this helpful comment regarding the variability at the 0.5 µg condition and the resulting statistical power. In the current chick dataset, focusing on the ventricular zone (VZ) in the images in Fig. 1C, ventricular surface coverage at 0.5 µg appears slightly lower than at 1 µg. However, because the coverage varies across embryos, this difference did not reach statistical significance in the present sample set. Nonetheless, the overall trend is that coverage is reduced at 0.5 µg relative to 1 µg. In the mouse experiments, although the readout was measured in differentiated neurons rather than VZ coverage, we likewise observed a difference between 1 µg and 0.5 µg, consistent with a reduction in labeling at the lower concentration. Achieving statistical significance for the chick VZ coverage comparison would require additional comparable samples; however, in this analysis we quantified only samples with similar GFP labeling efficiency, and therefore we were not able to further increase the n for Fig. 1C in the present revision.

We thank the reviewer again for this valuable suggestion.

7. Fig 1C-H: How was the number of clones determined? The brain slices show *iOnCAG[∞]RFP* positive cells but these must be a combination of electroporated cells and their derivatives. How can the number of clones be determined when the cells are labelled with a single colour?

Response: Thank you for this helpful comment. We only quantified clone numbers under the very sparse conditions (10 ng and 50 ng donor; i.e., 0.01–0.05 µg). In these settings, labeling was dilute enough that, within a single large FoV, we observed only a few, fully separated multicellular clusters. We counted clones only in images where putative clones were completely isolated; any samples in which clusters overlapped or intermingled were excluded a priori from clone counting.

At donor amounts ≥ 100 ng, such sparse labeling was not achievable and, as the reviewer notes, with single-color $iOnCAG\infty RFP$ it is not possible to determine unambiguously whether labeled cells derive from one clone or multiple clones. Accordingly, we did not report clone number at these higher doses; instead, we summarized labeling using alternative metrics, such as the fractional fluorescent area / labeling density per FoV (and related intensity-based measures; see Methods), in place of clone counts.

8. *Fig 2D. It's not clear from the images what the combinations of expressed fluorophores are in the cells highlighted by the boxes.*

Response: We appreciate this helpful comment. In response, we performed additional experiments and replaced Fig. 2D with a clearer set of images. In the revised Fig. 2D, we now include the corresponding single-channel views (R/G/B) alongside the merged image, making it easier to identify the fluorophore combinations and to distinguish signals with different subcellular localizations. Please refer to the revised Fig. 2D.

We thank the reviewer for this constructive suggestion.

9. *Line 89: typo: labelling density in chick the chick cortex*

Response: Thank you for catching this typo. We have corrected it in the revised manuscript to “labelling density in the chick pallium.” (see page 3, line 103)

10. *Fig 3: The text and the legend refer to the mouse data and then the chick data, but the image is presented in the opposite order.*

Response: Thank you for pointing this out. We have swapped the order of “mouse” and “chick” in the text and the figure legend so that they now correspond correctly to the image order in Fig. 3.

11. *Fig 3B and D: The cell identifiers are overlaid, making them illegible.*

Response: Thank you for this helpful suggestion. As the cell identifier markers were obscuring the image and were not legible, we prepared versions without the markers and instead delineated each clone using lines for clarity (Fig. 3B, D). We greatly appreciate the reviewer’s comment.

12. *When the three-colour iOn switch system was used in the mouse cortex, how were projection neurons and glial cells distinguished?*

Response: Thank you for this question. In our mouse cortex datasets with the three-color iOn switch, our analyses cover E16–E18, i.e., up to the onset of gliogenesis. As such, we primarily relied on

morphology at these stages. As shown in the original Neuron paper (Fig. S5A) and in our Fig. 4B and 4C (rat and guinea pig), astrocytes and oligodendrocytes are readily distinguishable from projection neurons by their characteristic morphologies. In Fig. 4B and 4C, we could readily distinguish projection neurons from astrocytes in rat and guinea pig brains based on morphological features. When a more definitive assignment was required, we used a non-overlapping fluorescence channel (such as Alexa Fluor 647) for immunostaining with NeuN (neuronal marker), which allows clear discrimination between neurons and glial cells.

13. Fig 4A. Are there tiny arrowheads/triangles pointing at cells in this image? These are too small to distinguish, not annotated and confusing. Please either remove them or make them large enough to see.

Response: Thank you for pointing this out. We had annotated cells from the same clone with tiny arrowheads. As you correctly noted, at this size and resolution they are confusing, so we have removed them. We appreciate your suggestion.

14. Fig 5: The legend does not accurately describe the figure. It would be helpful to label ventricular zone and cortical plate on the images. For the enlarged images, which one is RFP and which one is GFP?

Response: Thank you for raising these points. We apologize for the insufficient annotations and inaccuracies in the legends. We have corrected the legends to accurately describe the figures, added labels for the ventricular zone (VZ) and cortical plate (CP) on the images, and annotated the enlarged panels to indicate which channel is RFP and which is GFP.

15. The authors state that the stability of the RFP signal “facilitated uninterrupted tracking of migrating cells from their origin to their destination (Fig. 5)”. Can they highlight/indicate RFP positive cells that are seen migrating in sequential timepoints to support this claim?

Response: Thank you for this constructive suggestion. In response, we have revised Figure 5 to highlight representative migratory neurons by adding annotations and red arrows that track the same RFP⁺ cells across time points (Fig. 5A), as well as a small trajectory plot. These additions visually demonstrate that individual neurons can be continuously followed from their point of origin to their final destination.

16. Fig 6: Why are there red cells clearly visible in the top merged channel in A which don't show in the red only channel below?

Response: Thank you for this comment. In the original figure, we generated the black-and-white (single-channel) panel using the same display settings as the top merged image and included it below to help readers identify the RFP signal more easily. However, we found that in the printed version (and in some viewing conditions), the black-and-white rendering could become harder to interpret, and we

were not able to resolve this visibility issue reliably. Therefore, in the revised figure we removed the black-and-white single-channel panel and now present the data using the color images only, which preserves visibility more consistently. We thank the reviewer for this helpful suggestion.

17. Please specify the control in 6D.

Response: Thank you for pointing this out. In Fig. 6D, we used CAG::RFP as the control. We have revised the in-figure labels (and the legend) to explicitly indicate “CAG::RFP (control)”.

18. Fig S1. Legend is incorrect for B and C.

Response: Thank you for pointing this out. We have corrected the Fig. S1 legend for panels B and C.

19. Fig S2. What are the arrowheads indicating?

Response: Thank you for pointing this out. The arrowheads indicate neurite-like cellular processes that are more readily observed with the ^{iOn}CAG∞RFP labeling compared to CAG::GFP. We apologize for the insufficient legend that caused confusion, and we have added an explicit explanation.

Methods:

20. Line 298: [INSERT ELECTROPORATOR MODEL].

Response: Thank you for pointing this out. We have revised the sentence to include the correct electroporator model.

21. The in utero electroporation method is confusing and contradictory. Were mice anesthetized via IP injection or isofluorane? This section needs rewriting for clarity.

Response: Thank you for pointing this out. We apologize for the confusion caused by inconsistent wording in the original Methods. Because the anesthesia regimens differ by species, we have rewritten this section for clarity and now describe the anesthesia procedures separately for each species (mouse, rat, and guinea pig), with full details added in the revised Methods (highlighted in red). This resolves the apparent contradiction in the previous description. (see in the method)

22. In ovo electroporation: If full methods are not provided, please at least provide references to the protocols used.

Response: Thank you for pointing this out. We apologize for the limited methodological detail. In the revised version, we have expanded the in ovo electroporation methods and added references (chick¹ and turtle²) to the protocols used.

Reviewers' comments:

In the following response letter, **comments from the reviewers and editors are presented in bold**, while *our responses are indicated in italic blue*, and **modification sentences in the revised manuscript are indicated in Red** text for clarity.

Reviewer #3 (Remarks to the Author):

This manuscript describes an update to the existing iOn labeling system (integration-coupled On genetic switch), initially described in 2020 (Kumamoto et al.), which is used for expressing multiple transgenes in random combinations across cell populations via genomic integration. The technique is useful for lineage analysis and circuit tracing, as well as assessment of gene function and other applications. One particularly useful application of this technique is labeling cell populations in multiple colors. One key advantage of the original iOn system was its ability to tie gene expression to genomic integration for multiple different transgenes in both culture systems as well as model organisms. The new version described here demonstrates several improvements: the authors have expanded the color palette; the technique now allows for subcellular localization of color expression (nucleus, membrane, or mitochondrial localization); the authors now incorporate instructions for titrating density of labeling; and the authors describe expanded accessibility for use across different species and non-model organisms. These are important improvements that are well described and make the technique more broadly applicable. There is no earth-shattering finding described in this manuscript, however.

We sincerely thank Reviewer 3 for the careful reading of our manuscript and the constructive, detailed feedback. The comments were highly valuable and helped us improve the clarity, rigor, and presentation of the study. We have addressed each point below and incorporated the corresponding revisions in the manuscript.

Figure 1 -

1a. The ability to titrate density is a nice touch that is clearly demonstrated by this manuscript. How many times were the experiments in Figure 1 repeated? There is no sample size described.

Response: Thank you for pointing this out. We have added the sample size information to the Figure Legends. Specifically, for Figure 1A–B the experiments were performed twice, with three transfected samples per condition in each repeat (2 repeats × 3 samples = n=6 per condition). We also specified the number of brains analyzed for each condition in Fig. 1C–E and Fig. 1G–H in the revised figure legends (see **Figure 1 legend**).

In addition, for Fig. 1C–F, although we performed electroporations multiple times, we applied a predefined quality-control (inclusion) criterion to ensure fair normalization by GFP transfection efficiency. Specifically, we included only samples with comparable GFP labeling levels for quantification, thereby minimizing variability attributable to differences in electroporation efficiency

rather than the titration condition itself. We have added the resulting final n values to the revised figure legends as follows: Chick→1 µg: n = 4; 0.5 µg: n = 3; 0.05 µg: n = 2; 0.01 µg: n = 5; Mouse→1 µg: n = 3; 0.5 µg: n = 3; 0.1 µg: n = 1; 0.05 µg: n = 3; 0.01 µg: n = 3.

We thank the reviewer for this constructive suggestion.

1b. How was labeling density determined/estimated? This is challenging to estimate in a field of unlabeled cells. If it was only estimated, this should be honestly reported.

Response: Thank you for highlighting this point, and we apologize for the lack of clarity. We have now clarified in the Methods and Figure Legends how labeling density/coverage was determined.

For Fig. 1D, labeling was assessed as an estimated labeling coverage in the ventricular zone, calculated from the fraction of the ventricular-zone area occupied by *iOn*CAG[∞]RFP-positive signal within the field of view (i.e., an area-based proxy).

For Fig. 1E (and related quantifications), labeling was quantified within the electroporated population as the ratio of *iOn*CAG[∞]RFP-positive cells to CAG::*GFP*-positive cells (RFP⁺/GFP⁺). Because non-electroporated cells are unlabeled and therefore not visible in our fields, this metric reports the relative fraction of *iOn*-labeled cells within the electroporated population, rather than the absolute fraction of all cells in the tissue. We have updated the Methods and the relevant figure legends accordingly (annotated in the revised Fig. 1D/E legends).

1c. Similarly, how were the number of clones determined? The color palette is quite limited in these experiments and makes it almost impossible to calculate/distinguish individual clones.

Response: Thank you for raising this important point, and we apologize for the lack of clarity. We agree that the limited color palette makes definitive clone identification challenging. In this study we adopted a conservative operational definition for counting: one contiguous multicellular cluster (>3 labeled cells) was treated as one putative clone. Single cells and two-cell clusters were not counted as clones. We have added this definition to the Figure Legends and clarified in the text that these counts provide a conservative estimate rather than absolute clone numbers (see Figure 1 legend).

1d. The descriptions of 1 ug should be clarified to describe 1 ug per embryo (if correct), and at least in the methods, details should be given regarding what volume this entailed for microinjections and what other factors were included in the injection mix. Very slim details here.

Response: Thank you for pointing this out, and we apologize for the ambiguity. In the revised manuscript we clarify that “1 µg” refers to the DNA concentration in the injection solution (1 µg/µL). We also now specify the injection volume (1 µL per embryo), which corresponds to a total of 1 µg DNA per embryo (except for the titration assay in Fig. 1C and F, 10 - 500ng), along with the composition of the

injection mix and plasmid ratios to the Methods, and inserted a clarifying line in the Figure Legends. (see page12, line419-420)

1e. What promoters were used to achieve expression in the different species?

Response: Thank you for raising this point. We used CAG promoter for all animal models, except for *pUbb* (zebrafish ubiquitin B promoter) promoter to the zebrafish embryos.

1f. From the figures it is clear that increasing concentration of DNA increases the labeling density; however, how repeatable are these specific concentrations? How many times were each of these experiments repeated? There is presumably variability in expression in such experiments.

Response: Thank you for raising this point. We agree that the repeatability and sample sizes for each DNA concentration should be explicitly stated. We have now added the number of independent repeats and biological replicates for each concentration/condition to the revised Figure Legends (and clarified how replicate-level values were computed). We apologize for the omission in the original submission. For example, in Fig. 1A–B the experiments were performed twice with three independent transfections per condition in each repeat (2 repeats × 3 transfections = n = 6 per condition). For the in vivo concentration series, each embryo/brain was treated as one biological replicate, and multiple FoVs within an embryo were averaged to yield a single value per embryo; error bars summarize variability across biological replicates. Please see the revised figure legends for panel-specific n and replicate counts.

We thank the reviewer again for highlighting this issue.

Figure 2 -

2a. In Figure 2, panel D is described as “six colors”, which is misleading as it is only three colors, with each expressed color in two different subcellular compartments. The authors should clarify this each time it is referred to.

Response: Thank you for pointing this out. You are correct that describing panel D as “six colors” is incorrect; it is three fluorophores, each targeted to two subcellular compartments. We have revised the legends accordingly to “three-color iOn constructs with different localization tags”. (see page4, line136)

Figures 3/4 -

3a. Regarding Figures 3/4, the high density of labeling and limited color palette (particularly in Fig 4C) make it impossible to accurately/unequivocally identify daughter cells that are dispersed. While the panels do generally demonstrate how the labeling appears at the given ages, conclusions in the text about lineage should be removed.

For example: “we found that projection neurons derived from a common progenitor dispersed broadly across the cortex” - data is not provided to support this statement. “labeling neurons revealed clear and interpretable clone groupings” this statement is not supported by the figures and should be removed or significantly expanded with quantitative data.

Response: Thank you for this important comment. We agree that, particularly under high-density labeling with a limited color palette, unequivocal identification of dispersed daughter cells is challenging and lineage-level conclusions must be supported by appropriate quantitative evidence.

For Fig. 4, establishing conditions suitable for clonal analysis requires species-specific optimization of the delivery/analysis stages, DNA amount, and electroporation parameters. In the current study, we were not able to obtain sufficiently sparse, quantitative datasets to support rigorous clonal reconstruction in each species shown in Fig. 4. Importantly, the intent of this paragraph and figure was not to claim that definitive clonal lineage analysis can be performed across all species under the conditions tested, but rather to provide a demonstration that iOn enables multicolor labeling across diverse species. We apologize that our original phrasing may have led to misunderstanding, and we have revised the text accordingly to avoid overinterpretation. Accordingly, as the reviewer suggested, we have removed or substantially toned down statements that overstated lineage-level conclusions (e.g., broad dispersion from a common progenitor and “clear and interpretable clone groupings”) and revised the corresponding text to limit our conclusions to what is directly supported by the data.

For Fig. 3, by contrast, we have strengthened the evidence for clonal interpretation by increasing the number of samples and incorporating spatial information that enables conservative clone calling within the analyzed regions. Therefore, we did not change the main conclusion of Fig. 3, but clarified the operational definition and scope of clone identification and ensured that the text reflects the level of support provided by the data.

We appreciate the reviewer’s guidance, which has improved the accuracy and clarity of our presentation.

Figure 5 -

5a. In Figure 5, the large panel images at right are not labeled or described - assuming GFP is the lower panel? Panel letters need to be added to this figure.

Response: The upper panel of 5a is RFP and the lower panel is GFP. We apologize for the missing annotations. We have added the necessary in-figure channel labels and panel letters.

5b. In the graph at far right, to compare RFP/GFP, it may be better displayed as percent intensity as opposed to raw signal intensity values.

Thank you for the suggestion. In response to this comment (and related feedback from other reviewers), we revised the visualization to better convey that $iOnCAG^\infty RFP$ shows more uniform expression than $CAG::GFP$. In the revised Fig. 5B, we now quantify cell-by-cell RFP/GFP intensity

ratios and present these data as a violin plot, rather than showing raw signal intensities. This representation more clearly illustrates the more consistent expression driven by iOn. Please see the revised Fig. 5B.

We hope this fully addresses the reviewer's concern.

5c. Can the overexposure in Figure 5/SF2 be managed with adjusted acquisition settings, e.g., decreasing laser power, gain, etc?

Response: Thank you for this suggestion. Overexposure can indeed be mitigated by adjusting acquisition settings (e.g., reducing laser power and detector gain). In our images, the localized saturation near the VZ occurs only in the *CAG::GFP* channel; the *iOnCAG[∞]RFP* channel was acquired under non-saturating settings. When we reduce the GFP intensity/gain to avoid VZ saturation, GFP-positive cells in the CP become difficult to detect due to their comparatively weaker signal, which compromises visualization of the migrating-cell population. We have clarified this acquisition trade-off in the revised figure legend.

We thank the reviewer for this constructive comment, which has helped improve the clarity of our presentation.

Figure 7 -

7a. The numbers in the Figure 7 figure legend do not refer to any numbers provided in the corresponding figure. Suggest adding/using a clear panel lettering system instead.

Response: Thank you for pointing this out, and we apologize for the confusion. We have revised the legend to explicitly indicate which image corresponds to each panel in the figure, so that the numbering in the legend matches the corresponding panels unambiguously. Where applicable, we also added a clear panel lettering system to the figure and updated the legend accordingly (Fig. 6 in the revised figure; formerly Fig. 7).

Larger/more general questions:

8. The new species accessibility is a strength of the manuscript; however, it seems a missed opportunity to address its usage in zebrafish, which seems an obvious application for *in vivo* multicolor lineage analysis (e.g. Brockway et al., 2019; Furlan et al., 2017). Can the authors address potential applications in zebrafish?

Response: We thank the reviewer for this insightful suggestion. Because zebrafish is a species in which stable transgenic approaches are relatively straightforward, we had not initially prioritized testing the iOn switch in this model. As the reviewer notes, zebrafish offers clear advantages for *in vivo* multicolor lineage analysis. Motivated by this comment, we have now performed proof-of-principle experiments in zebrafish embryos and added these data to Fig. 4D and Fig. S5. Using microinjection

of *PBase* mRNA together with iOn constructs, we observed robust multicolor expression, with clear three-color labeling particularly in neural and muscle lineages. These new results demonstrate that the iOn switch is applicable in zebrafish and strengthen the cross-species scope of the manuscript.

We sincerely thank the reviewer for this suggestion, which helped improve the quality of our study.

(see page 5, line 163-169)

In zebrafish, multicolor labeling was readily detected in spinal neurons at 1 dpf (Fig. S5), while iOn switch expression became prominent in muscle after 5 dpf (Fig. 4D; Fig. S5). This tissue- and stage-dependent expression pattern in zebrafish may reflect differences in promoter usage (ubiquitin B, pUbb, in zebrafish versus pCAG in other species), species-dependent piggyBac integration efficiency, and/or the delivery method (microinjection versus electroporation). Clear GFP fluorescence was observed in more than 60% of injected animals. In approximately 10% of injected animals, muscle labeling was detected throughout the trunk, as shown in Fig. 4D.

9. If the authors wish to show that this is an improvement over previous iOn techniques, some sets of comparison images may strengthen this.

Thank you for this suggestion. In this revision, the key improvements over our previous iOn system are (i) the expanded color set and (ii) the addition of subcellular localization tags, which together increase combinatorial labeling capacity and facilitate downstream analyses. We summarize these design upgrades in Fig. 2. Rather than adding an extensive set of side-by-side images for all prior versions, we plan to present a concise overview comparing the previous and updated configurations in the graphical abstract, which highlights the specific improvements introduced here. We thank the reviewer again for this helpful comment.

10. In general, microinjecting increasing amounts of DNA into embryos often leads to toxicity effects. Can the authors discuss potential toxicity? This is particularly relevant for this work due to the multiple plasmids that are used for iOn.

Response: We thank the reviewer for raising this important point regarding potential toxicity associated with higher DNA doses and the use of multiple plasmids. We have added text to the Methods and Discussion addressing DNA load–related toxicity.

At the in vitro level, we previously reported that single iOn and LiOn plasmids do not measurably affect cell viability in HEK cells (Kumamoto et al., 2020, *Neuron*, Fig. S1G). In vivo, in utero electroporation under standard parameters is widely used and generally well tolerated^{3,4}. Nevertheless, we acknowledge that DNA delivery can elicit stress responses and apoptosis in some contexts⁵, that large plasmids can exacerbate electroporation-associated toxicity⁶, and that local cell death and microglial activation have been reported following in vivo electroporation in certain settings⁷. Accordingly, we used endotoxin-free, high-quality plasmid preparations⁸, limited per-plasmid concentration to ~1 µg/µL, constrained the total DNA load during co-electroporation, and optimized pulse parameters.

To directly evaluate potential cytotoxicity specifically associated with introducing multiple iOn plasmids, we performed additional experiments in chick embryos. We co-electroporated three iOn plasmids together with PBase at E4, and conducted time-lapse imaging from E7. Under these conditions, ~98% of labeled cells migrated without signs of cell death. In addition, on E7 fixed sections we performed cleaved caspase-3 immunostaining and observed no increase in caspase-3–positive cells upon iOn electroporation (Fig. S4). Together, these results indicate that co-delivery of multiple iOn plasmids under our experimental conditions does not measurably increase cytotoxicity, alleviating this concern.

We hope this fully addresses the reviewer’s concern.

(see page 7, line 239-251)

One potential concern is cytotoxicity associated with in utero electroporation. Under standard parameters, in utero electroporation is generally reported to yield high embryo survival and low cytotoxicity^{20,21}. However, electroporation-associated toxicity can increase with plasmid size and/or total DNA load²². Accordingly, in this study we used endotoxin-free plasmid preparations²³, limited the concentration of each plasmid to ~1 µg/µL, minimized the total DNA amount during co-electroporation^{21,24}, and optimized pulse settings. To directly assess cytotoxicity in our system, we performed (i) time-lapse imaging under high-concentration electroporation conditions and (ii) cleaved caspase-3 immunostaining on brains electroporated under the same conditions. In time-lapse recordings, we rarely observed cells undergoing cell death (Fig. S3A, and Movie 1), and cleaved caspase-3 signal was not increased in the electroporated region compared with surrounding tissue (Fig. S3Bb-EP region, compared to Ba-no-EP region). Together, these results suggest that the high-DNA electroporation used for multiplex clonal analysis with the iOn switch induces minimal cytotoxicity in our experimental conditions.

11. What are the expected coexpression levels for multiple injected constructs?

Response: We thank the reviewer for raising this important question regarding the expected level of coexpression when multiple constructs are delivered in vivo. To address this, we co-electroporated three iOn reporters (three colors) into the chick pallium at E4 and analyzed fixed brains at E7. We then generated ternary plots based on per-clone fluorescence ratios to visualize the relative contributions of the three reporters. In this representation, clones dominated by a single reporter cluster near the vertices, clones with contributions from two reporters distribute along the edges, and clones showing contributions from all three reporters occupy the central region (Fig. top panels).

In the representative example shown here, all six clones fall within the region consistent with contributions from three reporters (≥ 3 copy; Fig. bottom panels). Consistent with this, our main dataset in Fig. 3 likewise shows that the vast majority of clones are classified as ≥ 3 copy, while ≥ 2 copy clones are observed only rarely. Together, these data indicate that under our conditions (co-delivery of three iOn plasmids; total DNA 1 µg per embryo), most labeled clones coexpress multiple injected constructs, most frequently consistent with coexpression from all three reporters.

We appreciate the reviewer’s constructive comment, which prompted this additional analysis and improved the clarity of our manuscript.

12. The authors suggest that folding of the fluorophore may delay expression in this system. This seems a stretch, particularly since it is not supported by the RFP control images in Figure 6. If this is a real possibility, the authors should cite some evidence for this conjecture.

Response: Thank you for this comment. In Fig. S5 (formerly Fig. 6), the control condition uses episomal *CAG::RFP*, for which transcription and translation can begin immediately after delivery; therefore, RFP can be detected at earlier time points than in the iOn condition. By contrast, iOn reporter expression requires successful genomic integration first, followed by transcription and translation, which naturally introduces an additional delay.

In this context, we mentioned as an additional possibility that, once protein production has started, accelerating fluorophore maturation could potentially improve the apparent onset of detectable fluorescence in the iOn condition. For example, using fluorophores with faster maturation kinetics, such as mScarlet-I3⁹, may help shorten the time to detectable signal even when using iOn. We agree that our original wording was unclear and that we did not provide supporting references. We have therefore revised the text to clarify that this is a speculative, optimization-oriented point rather than a definitive explanation for the observed delay, and we have added the appropriate reference in the revised manuscript. Thank you again for the helpful suggestion.

(see page 8, line 282-287)

Instead, other steps, such as the integration process and subsequent transcription/translation, may contribute more significantly to the delay. In addition, once the reporter protein is produced, fluorophore maturation kinetics could further influence the time to detectable fluorescence. Future iterations of the system could therefore explore fluorophores with faster maturation (e.g., mScarlet-I3)²⁶ or reporter designs with reduced maturation time to further shorten this window, which would be especially valuable for early lineage studies or short-term live imaging.

Smaller comments

13. It would be nice if the authors referred to the original iOn paper (ref 20) in the Introduction where it is first discussed. The reference does not appear until line 181.

Response: Thank you for pointing this out. In the revised manuscript, we have added a citation to the original iOn paper (ref. 20) at the first mention in the Introduction (line 73-75). Please see the updated text.

14. Typos are present in lines 89, 168, and 298.

Response: Thank you for pointing this out. We have reviewed the manuscript and corrected the typographical errors at lines 89, 168, and 298.

1. Wada, K. *et al.* Lineage analysis of pallial subdivision-derived cells in the developing chick pallium. *Dev. Growth Differ.* **67**, 466–478 (2025).
2. Nomura, T., Yamashita, W., Gotoh, H. & Ono, K. Genetic manipulation of reptilian embryos: toward an understanding of cortical development and evolution. *Front. Neurosci.* **9**, 45 (2015).
3. Saito, T. In vivo electroporation in the embryonic mouse central nervous system. *Nat. Protoc.* **1**, 1552–1558 (2006).
4. Pacary, E. *et al.* Visualization and genetic manipulation of dendrites and spines in the mouse cerebral cortex and hippocampus using in utero electroporation. *J. Vis. Exp.* (2012) doi:10.3791/4163.
5. Li, L. H. *et al.* Apoptosis induced by DNA uptake limits transfection efficiency. *Exp. Cell Res.* **253**, 541–550 (1999).
6. Lesueur, L. L., Mir, L. M. & André, F. M. Overcoming the specific toxicity of large plasmids electrotransfer in primary cells in vitro. *Mol. Ther. Nucleic Acids* **5**, e291 (2016).
7. Rosin, J. M. & Kurrasch, D. M. In utero electroporation induces cell death and alters embryonic microglia morphology and expression signatures in the developing hypothalamus. *J. Neuroinflammation* **15**, 181 (2018).
8. Mamat, U. *et al.* Detoxifying *Escherichia coli* for endotoxin-free production of recombinant

proteins. *Microb. Cell Fact.* **14**, 57 (2015).

9. Gadella, T. W. J., Jr *et al.* mScarlet3: a brilliant and fast-maturing red fluorescent protein. *Nat. Methods* **20**, 541–545 (2023).